# MITOL deficiency triggers hematopoietic stem cell apoptosis via ER stress response

Wenjuan Ma [1,5], Shah Adil Ishtiyaq Ahmad[1,4,5], Michihiro Hashimoto [1,5], Ahad Khalilnezhad[1], Miho Kataoka[1], Yuichiro Arima[1], Yosuke Tanaka [1], Shigeru Yanagi [2], Terumasa Umemoto [1✉] & Toshio Suda [1,3✉]

## Abstract

**Hematopoietic stem cell (HSC) divisional fate and function are determined by cellular metabolism, yet the contribution of specific cellular organelles and metabolic pathways to blood maintenance and stress-induced responses in the bone marrow remains poorly understood. The outer mitochondrial membrane-localized E3 ubiquitin ligase MITOL/MARCHF5 (encoded by the *Mitol* gene) is known to regulate mitochondrial and endoplasmic reticulum (ER) interaction and to promote cell survival. Here, we investigated the functional involvement of MITOL in HSC maintenance by generating MX1-cre inducible *Mitol* knockout mice. MITOL deletion in the bone marrow resulted in HSC exhaustion and impairment of bone marrow reconstitution capability in vivo. Interestingly, MITOL loss did not induce major mitochondrial dysfunction in hematopoietic stem and progenitor cells. In contrast, MITOL deletion induced prolonged ER stress in HSCs, which triggered cellular apoptosis regulated by IRE1α. In line, dampening of ER stress signaling by IRE1α inhibitor KIRA6 partially rescued apoptosis of long-term-reconstituting HSC. In summary, our observations indicate that MITOL is a principal regulator of hematopoietic homeostasis and protects blood stem cells from cell death through its function in ER stress signaling.**

**Keywords** MITOL; Apoptosis; Cell Cycle; ER Stress Response; IRE1
**Subject Categories** Autophagy & Cell Death; Haematology; Stem Cells & Regenerative Medicine

## Introduction

Adult hematopoietic stem cells (HSCs) maintain quiescence within the bone marrow (BM) niche by having a low mitochondrial metabolism and low protein synthesis (Nakada et al, 2010; Signer et al, 2014; Nakamura-Ishizu et al, 2020). Once HSCs' steady state is switched to a cycling state by extrinsic or intrinsic stimuli, HSCs are programmed either to maintain self-renewal, give rise to variable circulating blood cells or go to apoptosis (Wilson and Trumpp, 2006). Correspondingly, cellular metabolic state alters appreciably through multiple crosstalk among organelles to guarantee energy homeostasis.

At first glance, as a major energy-producing organelle, mitochondria play a significant role in regulating the cellular metabolism of HSCs as well as governing their differential fate. Mitochondrial activity is directly linked to the determination of HSC cell fate and functioning as its increase in the form of oxidative phosphorylation leads to cell cycle activation, reactive oxygen species (ROS) production and eventual loss of long-term engraftment potential (Ito et al, 2019). Alternatively, mitochondria morphology also tightly controls HSCs fate to survival or cell death through fission and fusion processes via DRP1 and Mitofusin1/2 (MFN1/2) correspondingly (Luchsinger et al, 2016; Cho et al, 2019), which are located on the mitochondrial outer membrane (MOM). Mitochondrial outer membrane permeabilization (MOMP) highly triggers cellular intrinsic apoptosis regulated by BCL-2 family, such as BAK and BAX (Karbowski et al, 2002), which subsequently initiates downstream apoptotic signaling (McArthur et al, 2018).

In addition, mitochondria interact with different cellular organelles to determine cell survival and death. Endoplasmic reticulum (ER) is strictly coordinated with mitochondria (Copeland and Dalton, 1959) to fine-tune protein production, control intracellular calcium levels and regulate lipid metabolism via mitochondria-associated ER membranes (MAMs) (Marchi et al, 2014). The outer mitochondrial membrane-localized E3 ubiquitin ligase MITOL/MARCHF5 (encoded by the *Mitol* gene) is a MAMs protein (Yonashiro et al, 2006), which can interfere with both mitochondrial fission and fusion processes through targeting DRP1 and MFN2 respectively, for ubiquitin-mediated degradation (Karbowski et al, 2007; Sugiura et al, 2013). Once protein misfolding and/or degradation disorder happens, chronic or unresolved ER stress initiates ER-specific unfolded protein response (UPR) through three typical branches: (1) protein kinase RNA (PKR)-like ER kinase (PERK), which response to ER stress through

[1]International Research Center for Medical Sciences (IRCMS), Kumamoto University, Kumamoto 8620811, Japan. [2]Laboratory of Molecular Biochemistry, School of Life Sciences, Tokyo University of Pharmacy and Life Sciences, Hachioji, Tokyo, Japan. [3]Cancer Science Institute of Singapore, National University of Singapore; Centre for Translation Medicine, Singapore 117599, Singapore. [4]Present address: Department of Biotechnology and Genetic Engineering, Mawlana Bhashani Science and Technology University, Tangail, Bangladesh. [5]These authors contributed equally: Wenjuan Ma, Shah Adil Ishtiyaq Ahmad, Michihiro Hashimoto. ✉E-mail: umemoto@kumamoto-u.ac.jp; sudato@keio.jp; csits@nus.edu.sg

phosphorylation of eukaryotic translation initiation factor-2 (eIF2) subsequently favoring ATF4 to initiate transcriptional activity and trigger apoptosis, autophagy and redox balance (Harding et al, 2000); (2) ATF6, relevant to mitochondria metabolism (Thuerauf et al, 2007); and (3) Spliced X-box-binding protein 1 (XBP-1s), responding to oligomerization of IRE1α and initiate protein folding and lipogenesis (Hollien and Weissman, 2006). Of note, coordination of IRE1α signaling with pro-apoptotic factors plays an important role in regulating ER stress-induced apoptosis in HSCs (Van Galen et al, 2014; Liu et al, 2019).

Reducing ER stress level by gene modification or the addition of chemical chaperones has been shown to improve HSC function (Miharada et al, 2014). Moreover, MITOL has been reported to regulate the cellular ER stress response by IRE1 ubiquitination, as the loss of ubiquitination leads to apoptosis (Takeda et al, 2019). Nevertheless, the effect of Mitol deletion on HSC function remains unknown. The unique position of MITOL at the crossroad between mitochondria and ER function makes it an interesting candidate to deeply explore the influence of organelles on HSC functioning in the context of Mitol deletion.

Here, we found that Mitol deletion in the BM leads to HSCs exhaustion and impairment of BM reconstitution capability and failed to maintain hematopoietic homeostasis. Interestingly, we could not detect major mitochondrial dysfunction or dysregulation in Mitol-deleted hematopoietic stem cells and progenitor cells (HSPCs), while Mitol deletion enhanced mitochondrial respiration. Furthermore, Mitol deletion induced ER stress and increased cellular apoptosis in HSCs, which was essentially mediated through IRE1α signaling. Inhibition of ER stress by the IRE1α inhibitor can partially rescue the apoptotic phenotype in HSCs. Our observations indicate that MITOL protects HSCs from apoptosis and maintains hematopoietic homeostasis through ER function, which provide references for MITOL function in healthy HSCs biology.

# Results

## Mitol deletion causes HSCs exhaustion

To study the role of MITOL in HSCs regulation or hematopoiesis, Mitol^Flox/Flox mice were crossed with MX1-cre +/− mice to obtain Mitol^Flox/Flox; MX1-cre +/− mice, which induces the deletion of Mitol in the hematopoietic lineage following an injection with Polyinosinic-polycytidylic acid (Poly:IC). Firstly, we analyzed the BM cells of Mitol^Flox/Flox; MX1-cre+ (thereafter, Mitol^Δ/Δ) mice at 4 weeks after Poly:IC induction. We observed that the number of BM nucleated cells was significantly reduced in Mitol^Δ/Δ mice compared to that in Mitol^Flox/Flox; MX1-cre- control mice (Fig. 1A). Moreover, the number of white blood cells (WBC) and platelets (PLT) within peripheral blood (PB) was greatly reduced in Mitol^Δ/Δ mice, while red blood cells (RBC) were not so different (Fig. 1B). Furthermore, Mitol deletion significantly decreased the frequency of long-term HSCs (LT-HSCs, Lin⁻CD117⁺EPCR⁺CD48⁻CD150⁺), short-term HSCs (ST-HSCs, Lin⁻CD117⁺EPCR⁺CD48⁻CD150⁻) and LEK (Lin⁻CD117⁺EPCR⁺) fractions within BM (Fig. 1C,D). Surprisingly, at earlier stage after the deletion of Mitol allele (1 week after Poly:IC injection), the frequency of LT-HSC and ST-HSC population was already reduced in Mitol^Δ/Δ mice (Fig. 1E,F), while the frequency of multipotent progenitors (MPPs;

Lin⁻CD117⁺EPCR⁺CD48⁺) and LEK cells was significantly enhanced (Fig. 1F). However, PB showed the slight reduction of RBC counts in Mitol^Δ/Δ mice, while WBC and platelets counts were still comparable between control and Mitol^Δ/Δ mice (Fig. EV1A). Overall, these data indicate that Mitol deletion induces BM failure accompanied by rapid exhaustion of HSCs.

## Mitol deletion compromises BM reconstitution engraftment and the maintenance of hematopoietic homeostasis

To assess the impact of Mitol deletion on HSC functions, we performed a competitive transplantation assay by using control or Mitol^Δ/Δ BM cells at 1 week after Poly:IC injection (Fig. 2A). Mitol^Δ/Δ BM cells showed significantly lower chimerism (less than 5%) within PB as well as LSK (Lin⁻CD117⁺Sca1⁺) or LT-HSC fraction of BM after the transplantation, compared to control BM cells (Fig. 2B,C), indicating the failure of BM engraftment by Mitol deletion. The similar negative effect of Mitol deletion on the chimerism within PB was observed when Mitol deletion was induced in BM-chimera mice generated by using Mitol^Flox/Flox BM cells (Fig. 2D,E). After 1 month post Mitol deletion, lineage distribution in PB sightly skewed towards T-cell lineage (Fig. 2F). Moreover, Mitol deletion did not affect the chimerism of LSK and LT-HSC fraction within BM at 1 month after Mitol deletion but was severely reduced at 3 months post transplantation (Fig. 2G). Taken together, these data suggest that Mitol deletion impairs the reconstitution of hematopoiesis after transplantation as well as maintenance of normal hematopoiesis through the exhaustion of HSPCs even under homeostatic conditions.

## Mitol deletion induces ER stress in HSCs

To clarify the mechanism of how Mitol deletion led to the failure of HSC maintenance, we compared gene expression pattern between control and Mitol^Δ/Δ HSCs by using bulk RNA-seq post 1-week Poly:IC injection. Principal component analysis (PCA) of RNA-seq data clearly distinguished between control and Mitol^Δ/Δ HSCs (Fig. EV1B). Unsupervised GO analysis with all DEGs (Fig. 3A, genes are listed in Fig. EV1C) revealed that cell cycle-related pathways were enormously upregulated in Mitol^Δ/Δ HSCs (Fig. 3B). Therefore, we checked the cell cycle status of Mitol^Δ/Δ HSCs by staining for the cell proliferation marker Ki67. Consistent with gene expression profile, Mitol^Δ/Δ HSCs showed a significantly reduced frequency of cells in G0 phase compared to the control (Fig. EV1D,E), while the frequency of cells in G1 phase was significantly increased in Mitol^Δ/Δ HSCs (Fig. EV1D,E). These suggested that Mitol^Δ/Δ HSCs fail to maintain their quiescence, which is involved in HSC exhaustion.

Furthermore, pathway analysis showed that Mitol deletion enhanced expression of genes related to ER stress response, intrinsic apoptosis and regulation of mitochondrial membrane permeability as well as mitochondrial depolarization (Figs. 3C and EV3A,B), suggesting that Mitol deletion induces HSCs apoptosis. Indeed, MITOL, as a MAMs protein (Yonashiro et al, 2006), functions as a regulator to coordinate mitochondria with ER to fine-tune protein production and control metabolic status (Marchi et al, 2014; Nakamura-Ishizu et al, 2020). Since Mitol deletion posed ER stress response as the top affected pathways (Fig. 3C–E,

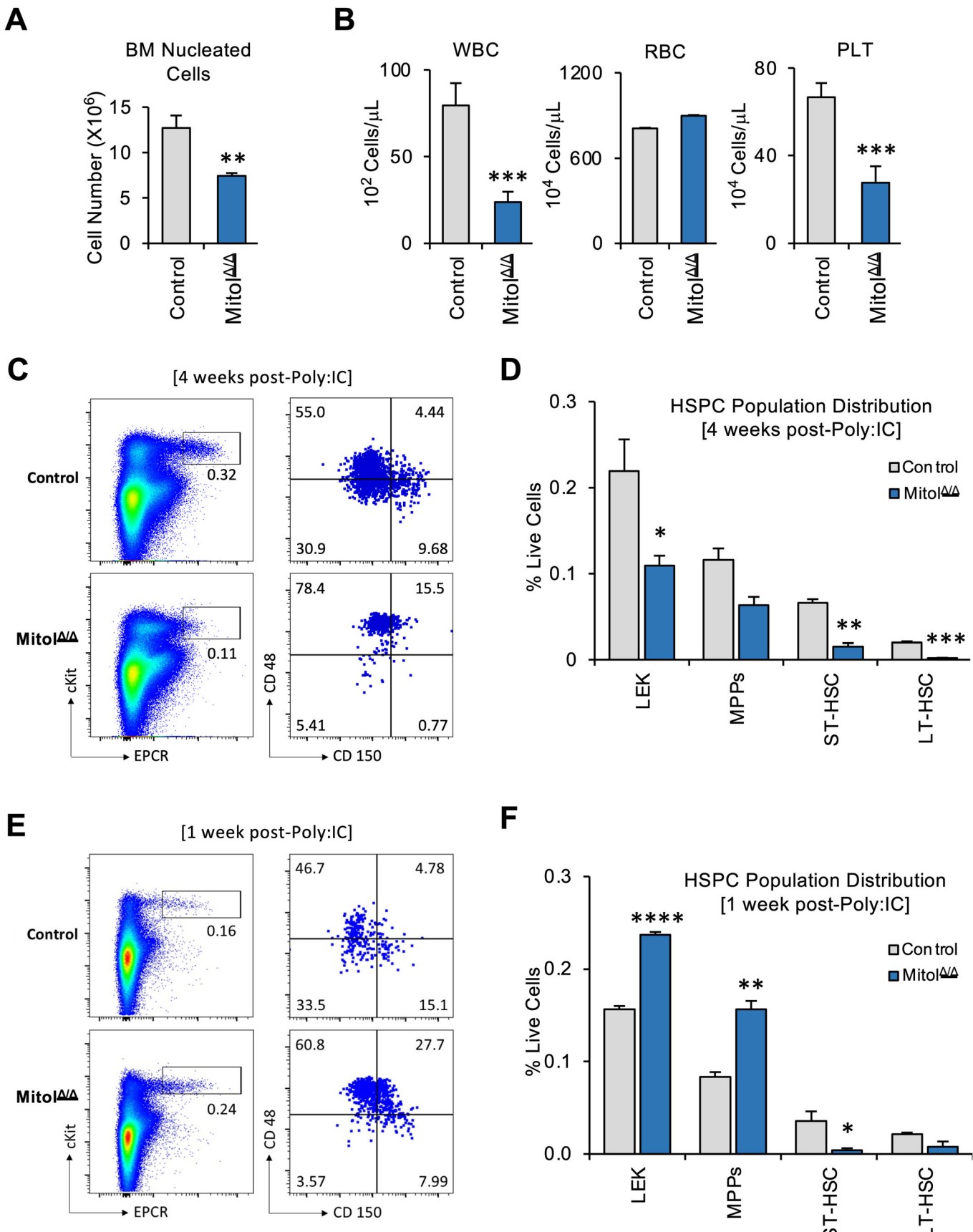

◄ **Figure 1.   Mitol deletion in the bone marrow leads to hematopoietic stem cell exhaustion.**

(A) Total nucleated BM cell count and (B) peripheral blood populations after 4 weeks of Poly:IC induced Mitol deletion ($n = 3$). (C, D) HSPC population distribution after 4 weeks of Poly:IC induction, shown in FACS plots (C) and graph (D) ($n = 4$). (E, F) HSPC population distribution after 1 week of Poly:IC induction, shown in FACS plots (E) and graph (F) ($n = 3$). Data Information: Data represent mean ± SEM with two-tailed unpaired Student's $t$ test. ns, $P > 0.05$; *$P < 0.05$; **$P < 0.01$; ***$P < 0.001$; ****$P < 0.0001$. Source data are available online for this figure.

genes are listed in Fig. EV2), we first focused on ER stress as a negative regulator for HSC maintenance in Mitol$^{\Delta/\Delta}$ mice. Correspondingly, Mitol deletion led to upregulated expression of spliced *Xbp1* and its target genes such as *Sec61b* and *Sec61a1* (Fig. 3F) (Hetz et al, 2011), suggesting activated IRE1α pathway, which is one of the major ER stress responses. Moreover, Mitol deletion elevated expression of *Atf6*, which is a central molecule in another major ER stress response pathway (Fig. 3F). In addition, *Bax*, *Bak1*, *Bok*, and *Trp53* expression levels were also significantly upregulated in ER-regulated apoptotic processes (Fig. 3F,I). Although CHOP, encoded by the *Ddit3* gene and a downstream molecule of ATF6 pathways for UPR, showed similar RNA expression levels between Mitol$^{\Delta/\Delta}$ and control HSCs (Fig. EV3C), the target gene *Bcl2l11* (also known as *Bim*) is positively upregulated (Fig. 3I). These data suggested that Mitol deletion enhanced ER stress response.

In addition, BAX/BAK, which is central members for IRE1α signaling (Cheng et al, 2001; Hetz et al, 2006), also acts as an essential regulator of mitochondrial permeabilization (Desagher and Martinou, 2000; Reimertz et al, 2003). Our data showed that Mitol deletion enhanced expression of genes related to mitochondrial permeabilization and depolarization in HSCs (Fig. 3G–I, gene list is shown in Fig. EV2), which might suggest mitochondrial regulation in Mitol$^{\Delta/\Delta}$ HSCs. However, gene expression pattern showed that Mitol deletion did not induce severe mitochondrial dysregulation or mitophagy (Fig. EV3D,E). These data suggested that Mitol deletion induced apoptosis in HSPCs through enhancing ER stress response but not through mitochondrial dysfunction.

### Mitol deletion does not induce mitochondrial dysfunction

To examine whether the mitochondrial function was impaired when Mitol deletion induced hematopoiesis failure, we focused on the metabolic status in HSCs, since MITOL is located on the outer membrane of mitochondria and involved in the regulation of mitochondrial fission and fusion (Shiiba et al, 2020). Intriguingly, after 1 week post Poly:IC induction, there was no significant difference between control and Mitol$^{\Delta/\Delta}$ cells in both mitochondrial parameters (Fig. 4A–D). Moreover, the potential for glucose uptake was also little difference between control and Mitol$^{\Delta/\Delta}$ HSCs (Fig. 4E). However, Mitol deletion led to significantly increased mitochondrial respiration (OCR: oxygen consumption rate) as well as glycolysis activity (ECAR: extracellular acidification rate) at baseline (Fig. 4F,G). This increment of OCR at baseline was due to both enhanced proton leak and ATP production (Fig. 4H). However, Mitol deletion did not affect maximal respiration, resulting in decreased spare respiratory capacity in Mitol$^{\Delta/\Delta}$ LSK cells (Fig. 4H). Overall, these results suggested that Mitol$^{\Delta/\Delta}$ HSPCs showed an enhanced basal mitochondrial metabolism possibly through increasing glycolysis without relying on significantly

enhanced glucose uptake, resulting in elevated proton leak and ATP production, but little mitochondrial dysfunction.

### Mitol deletion induces ER stress-mediated apoptosis in HSCs

Next, to confirm whether Mitol deletion induces enhanced cell death in HSCs, we first cultured control or Mitol$^{\Delta/\Delta}$ HSCs for 1 week. Surprisingly, Mitol deletion significantly reduced total viable cell number (Fig. 5A), even though Mitol$^{\Delta/\Delta}$ HSCs have enhanced expression levels of genes related to cell cycle (Fig. 3B). In addition, after in vitro culture (36 h), Mitol deletion significantly enhanced the frequency of Annexin V$^+$ apoptotic cells (Fig. 5B,C). These data suggested that Mitol$^{\Delta/\Delta}$ HSCs showed decreased proliferation due to apoptosis. To validate the possibility that Mitol deletion enhanced ER stress response, we assessed the effect of Mitol deletion on the protein level of spliced XBP1 (XBP-1s) in HSCs, which is one of the hallmarks of IRE1α-mediated ER stress (Hollien and Weissman, 2006). Moreover, IRE1α is identified as a substrate for MITOL (Takeda et al, 2019). Results showed that Mitol deletion significantly enhanced the level of XBP-1s in HSCs (Fig. 5D,E). Moreover, the phosphorylation level of eIF2α, a central member of PERK pathway for ER stress response, was also elevated in Mitol$^{\Delta/\Delta}$ HSCs (Fig. 5D,E). Furthermore, Mitol deletion showed higher expression of ATF4 protein level (Fig. 5F) and Xbp-1s mRNA level (Fig. 5G) in HSCs. These results support that IRE1a-XBP1 pathway contributes to UPR/ER stress in Mitol$^{\Delta/\Delta}$ HSCs. In addition, consistent with RNA-seq data, upregulated expression levels of *Bax* and *Bak1* in Mitol$^{\Delta/\Delta}$ HSCs were validated by RT-qPCR (Fig. EV3F). These data suggest that Mitol deletion causes ER stress-induced apoptosis in HSCs. Finally, to examine whether IRE1α-mediated ER stress is involved in the induction of apoptosis in Mitol$^{\Delta/\Delta}$ HSCs, we cultured control or Mitol$^{\Delta/\Delta}$ LT-HSCs in the absence or presence of KIRA6, an IRE1α kinase inhibitor (Van Galen et al, 2014; Liu et al, 2019). Interestingly, KIRA6 treatment partially canceled the effect of Mitol deletion as an increased frequency of Annexin V$^+$ cells after the culture of HSCs was observed (Fig. 6A,B). These data indicated that ER stress response mediated by IRE1α signaling partially contributes to cellular apoptosis caused by Mitol deletion (Fig. 6C).

## Discussion

Generally, the interaction between organelles such as mitochondria and ER is crucial for several biological events in terms of cellular regulations (Nakamura-Ishizu et al, 2020). In this study, we evaluated the role of MITOL, which is localized at the mitochondria and an important regulator for both mitochondrial and ER function, by using specific deletion of *Mitol* in hematopoietic lineage (MX1-cre system). Indeed, MITOL contributes to HSC

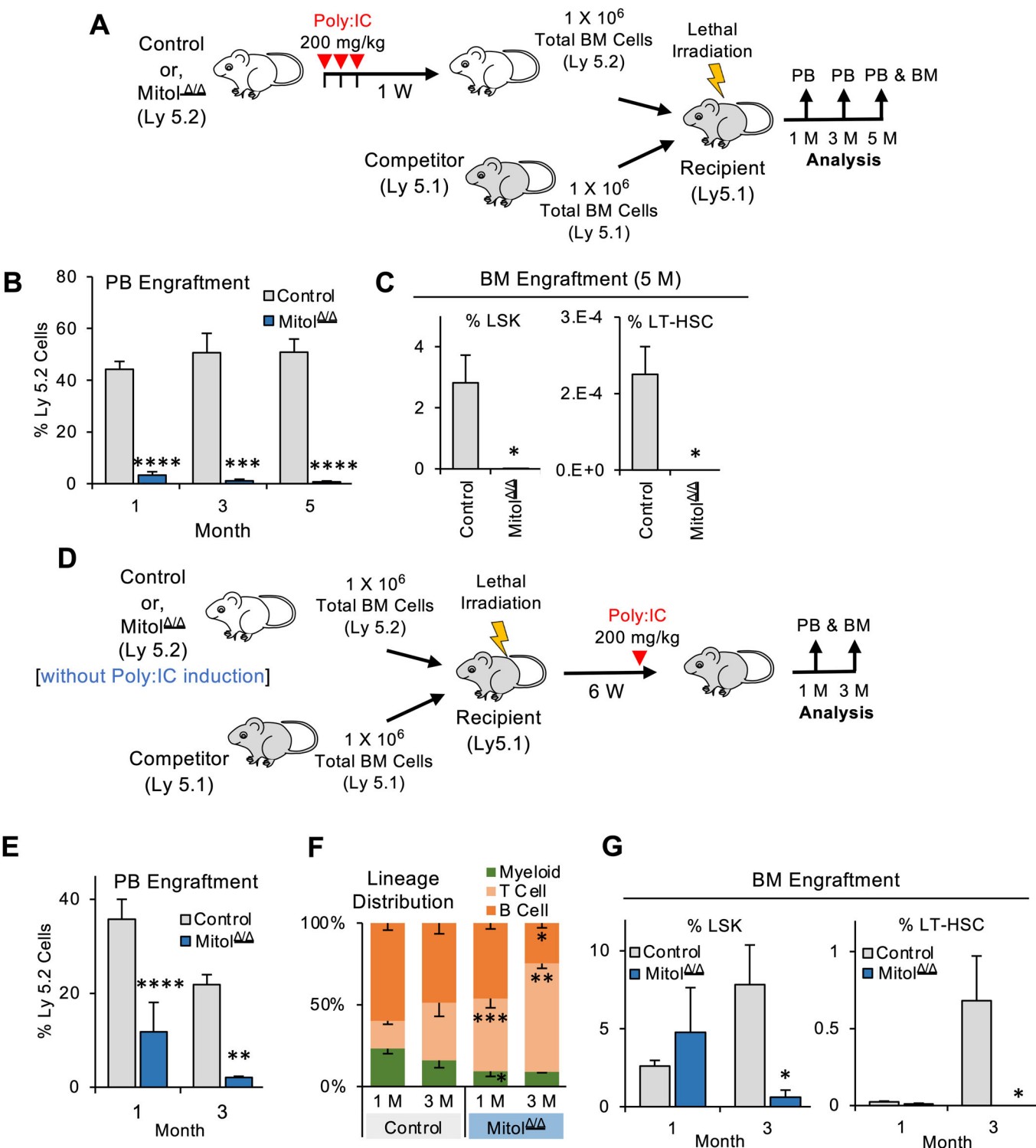

**Figure 2. Mitol deletion compromises BM engraftment and reconstitution potential.**

(A) Experiment strategy for BM transplantation for the data shown in (B, C). (B) PB engraftment followed through months 1, 3, and 5 after BM transplantation ($n = 10$). (C) BM engraftment after 5 months of BM transplantation ($n = 3$). (D) Experiment strategy for chimerism maintenance analysis for the data shown in (E–G). (E) PB engraftment followed through months 1 and 3 following Poly:IC injection of chimeric mice. (F) Lineage distribution in the reconstituted PB following Poly:IC injection of chimeric mice ($n = 10$). (G) BM engraftment after 1 and 3 months of Poly:IC induction ($n = 5$). Data Information: Data represent mean ± SEM with two-tailed unpaired Student's $t$ test. ns, $P > 0.05$; *$P < 0.05$; **$P < 0.01$; ***$P < 0.001$; ****$P < 0.0001$. Source data are available online for this figure.

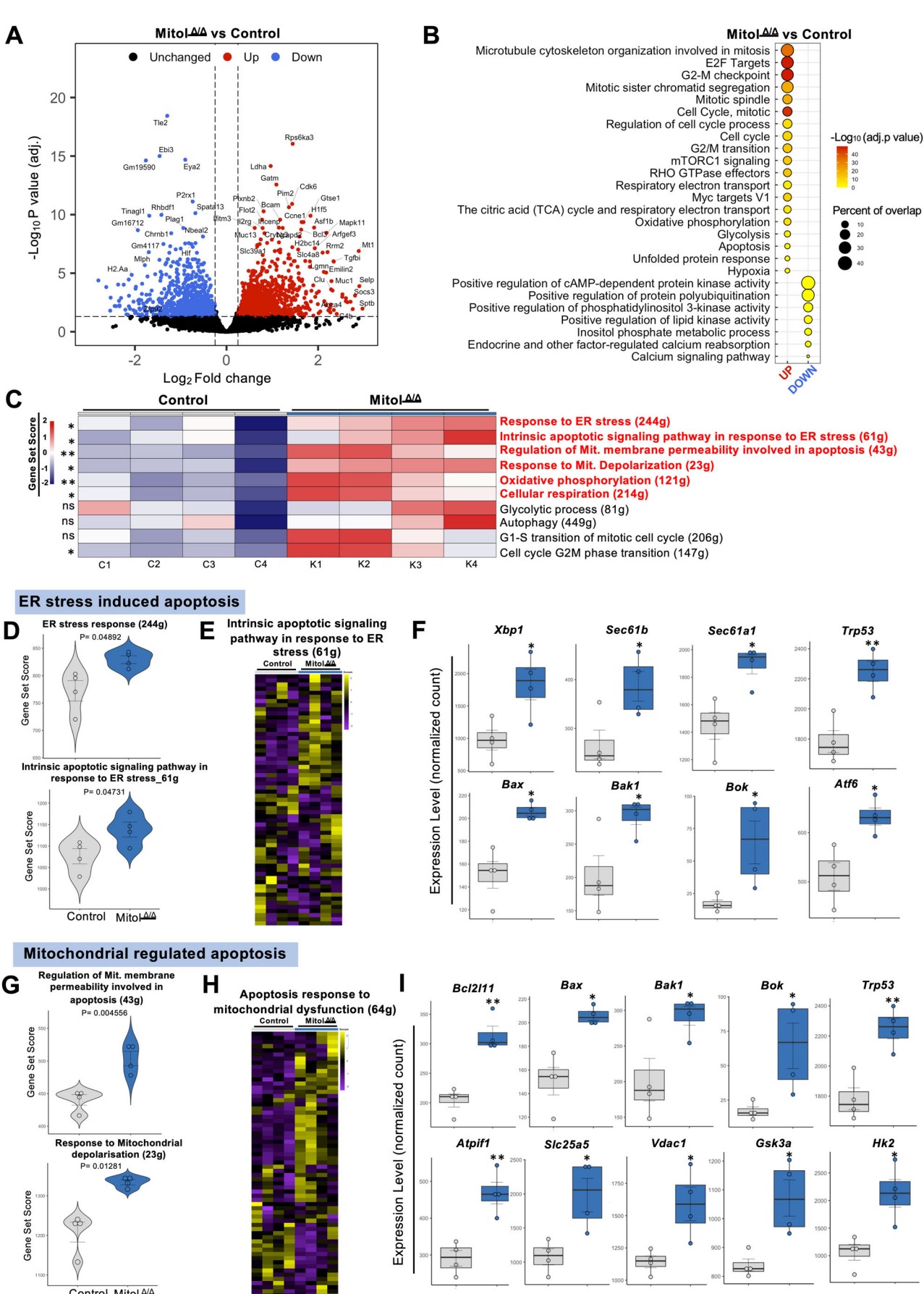

◄  **Figure 3.  Mitol deletion induces ER stress in HSCs.**

(A) Volcano plot of Top 100 DEGs from bulk RNA-seq. (B) Summary of GO terms significantly altered in the HSCs following Mitol deletion. (C) Pathway analysis of metabolic genes involved in ER and mitochondrial activity. (D–F) ER stress-induced apoptosis is shown in violin plots (D), heatmap (E), and box plots (F). (G–I) Mitochondrial-regulated apoptosis is shown in violin plots (G), heatmap (H) and box plots (I) ($n = 4$). Data Information: (D, G) horizontal lines in violin plots represent quartiles. (F, I) The box plot's central band marks the median, boxes mark the first and third quartiles, and whiskers extend the boxes to the largest value no further than 1.5 times the interquartile range. Data were analyzed with the Wald test. ns, $P > 0.05$; *$P < 0.05$; **$P < 0.01$. Source data are available online for this figure.

survival by preventing ER stress-induced apoptosis, which is required for the maintenance of hematopoietic homeostasis in steady state, as evidenced by BM failure in Mitol$^{\Delta/\Delta}$ mice (Figs. 1 and 2D–G). Although Mitol deletion quickly led to decreased HSC numbers, progenitor cells were simultaneously increased immediately after Poly:IC injection (Fig. 1E,F). These data suggest that Mitol deletion contributes to the differentiation of HSCs into progenitor cells. This possibility is supported by both cell cycle progression (Fig. EV1) and enhanced expression of genes related to cell differentiation (Fig. EV4). In addition, deletion alters lineage distribution towards T-cell differentiation, which is consistent with enhanced expression of genes relevant to differentiation pattern into T cells (Figs. 2 and EV4). Moreover, decreased myeloid cells and B cells in Mitol$^{\Delta/\Delta}$ mice suggested that Mitol deletion might be crucial for myeloid differentiation and B cell maturation. Therefore, MITOL may also be involved in the regulation of differentiation and/or lineage commitment in HSCs (Fig. EV4). However, Mitol deletion led to BM failure, which greatly affects the environment surrounding HSCs. Therefore, it remains unclear whether the effect of Mitol deletion on differentiation in HSCs was cell-autonomous.

Regarding the effect of MITOL deficiency on glycolysis, we observed that Mitol$^{\Delta/\Delta}$ HSCs showed comparable glucose uptake with control HSCs (Fig. 4E), which is consistent with gene expression pattern related to glycolysis (Fig. EV3A). However, Mitol deletion enhanced ECAR in HSPCs (Fig. 4G). These results may suggest that Mitol deletion enhances glycolysis without enhanced glucose uptake, but the mechanism is still unknown. Similarly, Mitol deletion hardly affected mitochondrial status in HSCs (Fig. 4A–D), but enhanced mitochondrial basal respiration in HSPCs (Fig. 4H). This enhanced mitochondrial basal respiration in Mitol$^{\Delta/\Delta}$ HSPCs was consistent with enhanced expression of genes involved in oxidative phosphorylation and respiratory ETC (Fig. EV5A,B). Therefore, MITOL deficiency may have the potential to enhance mitochondrial metabolism. Since cell cycle progression was simultaneously observed in Mitol$^{\Delta/\Delta}$ HSCs (Figs. 3B and EV1D,E), both mitochondrial basal respiration and EACR might be enhanced in response to cell cycle progression. It is well known that MITOL can interfere with both mitochondrial fission and fusion process through the targeting of DRP1 and MFN2 respectively via ubiquitination (Cho et al, 2019; Luchsinger et al, 2016). Although the possibility that Mitol deletion affects these processes cannot be completely excluded, we concluded that although MITOL generally behaves as the regulator for mitochondrial fission and fusion, that is not the case in HSCs where MITOL seems to exert less of its regulatory effects on mitochondrial function.

However, Mitol deletion upregulated ER stress-mediated apoptosis in HSCs (Fig. 3), which may be one possible explanation of BM failure in Mitol$^{\Delta/\Delta}$ mice (Fig. 2). Although MITOL resides on the outer membrane of mitochondria, it was reported to restrict ER stress through the ubiquitination of IRE1α (Takeda et al, 2019). Indeed, our data suggest that MITOL is involved in the regulation of ER stress-mediated apoptosis through IRE1α-XBP-1s axis (Figs. 5

and 6). However, the mechanism of how ER stress was initiated in Mitol$^{\Delta/\Delta}$ HSCs is still unclear. Given the suppressive effect of MITOL on ER stress response through ubiquitinating IRE1α, Mitol deletion may lead to the accumulation of IRE1α, thereby amplifying ER stress response or enhancing the sensitivity against unfolded or misfolded proteins (Mohrin et al, 2015; Senft and Ronai, 2015). After the transplantation of BM cells, the chimerism of Mitol$^{\Delta/\Delta}$ cells within PB was largely decreased compared with that of control cells (Fig. 2), but Mitol deletion after generating Mitol$^{Flox/Flox}$ BM-chimera mice exerted milder effect on the chimerism of Mitol$^{\Delta/\Delta}$ cells. Since protein synthesis is generally promoted under proliferative conditions such as after transplantation (Signer et al, 2014), this greater effect of Mitol deletion after the transplantation may support the possibility that Mitol deletion further enhanced ER stress response. Therefore, MITOL seems to be essential to appropriately regulate ER stress response in HSCs, probably maintaining hematopoietic homeostasis. Definitely, other studies have reported the potential of MITOL function in neurodegenerative diseases and aging (Yonashiro et al, 2009, 2012; Park et al, 2010), which support the hypothesis that MITOL contributes to the maintenance of homeostasis in tissues through preventing cell death and aging. However, since the regulatory mechanism(s) of MITOL expression or activation in HSCs remain unclear, clarifying this will help to better understand the role of interaction between ER and mitochondria during the maintenance of HSCs or hematopoietic homeostasis under physiological conditions.

There are three major pathways upon response to ER stress and sensing UPR, which are IRE1α, PERK and ATF6 pathways (Kadowaki and Nishitoh, 2013). Our data showed that the inhibition of IRE1α partially reduced Mitol deletion-induced apoptosis, suggesting the possibility that the other two pathways may also contribute to Mitol deletion-induced apoptosis through ER stress response. To date, PERK signaling was recently demonstrated to be essential for reprogramming HSPCs to commit to myeloid-derived suppressor cells (MDSC) differentiation (Liu et al, 2022). In addition, dysfunction of ATF6 was reported to loss the ER quality control in mammalian cells (Yamamoto et al, 2007). However, the precise molecule triggering these three pathways on HSCs still remains unclear. In addition, BAX and BAK, proteins involved in mitochondrial permeabilization, are associated with IRE1 signaling to facilitate cellular apoptosis (Hetz et al, 2006) in a CHOP-dependent manner (Lin et al, 2007; Cheng et al, 2001). Although Mitol deletion did not induce major impairment of mitochondrial function, it is still unclear whether BAX/BAK can coordinate the downstream pathways between ER and mitochondria to regulate cell fate upon Mitol deletion in HSCs (Fig. EV5C,D).

In this study, we found that MITOL is essential for HSC survival and maintenance of hematopoietic homeostasis through regulating ER stress. It is interesting that Mitol deletion affects ER functions more so than the mitochondria, despite the residential advantage of

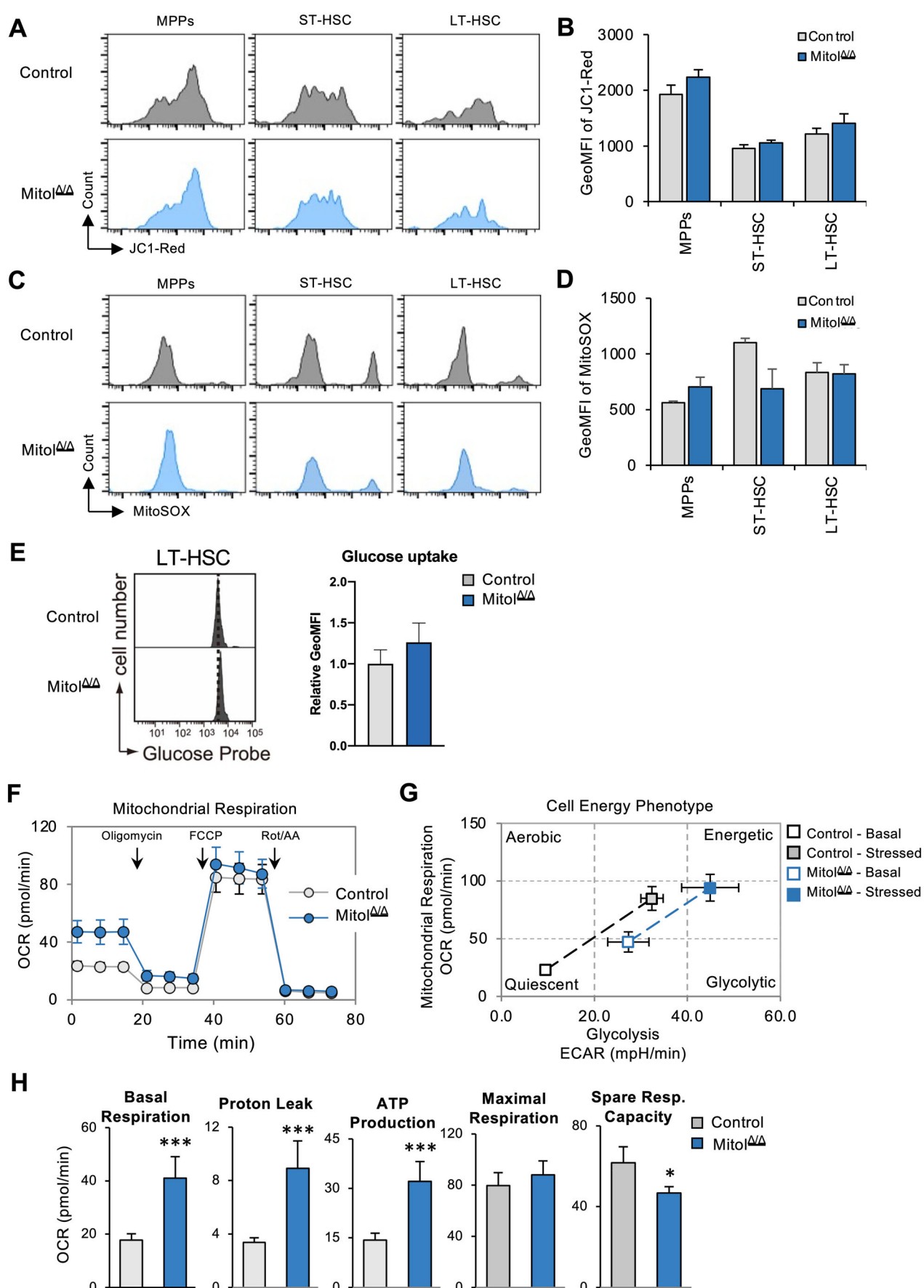

**Figure 4. Mitol deletion enhances mitochondrial respiration of HSPCs without major dysfunction.**

(A, B) Analysis of mitochondrial membrane potential of the HSPCS after 1 week of Poly:IC injection by JC-1-Red staining. A representative histogram of FACS staining is shown in (A) and the graph is shown in (B) ($n = 3$). (C, D) Mitochondrial ROS production by MitoSOX staining in the HSPCs 1 week post Poly:IC injection is shown by the histogram (C) and graph (D) ($n = 3$). (E) Glucose uptake in LT-HSC of Control and Mitol$^{\Delta/\Delta}$ 1 week of Poly:IC injection ($n = 3$). (F) In vivo measurement of mitochondrial respiration of LSK (Lin-Sca1+c-Kit + ) cells by Seahorse metabolic flux analyzer after 1 week of Poly:IC induction ($n = 3$). (G) Cell energy phenotype analysis by comparing mitochondrial respiration and glycolytic rate ($n = 3$). (H) Comparison of several mitochondrial metabolic parameters between Control and Mitol$^{\Delta/\Delta}$ LSK cells ($n = 3$). Data Information: Data represent mean ± SEM/SD with two-tailed unpaired Student's *t* test. ns, $P > 0.05$; *$P < 0.05$; ***$P < 0.001$. Source data are available online for this figure.

MITOL on the mitochondrial outer membrane. Our findings provided an example of how organelle interactions such as the association between ER and mitochondria controls HSC maintenance. Recently, there are increasing reports regarding organelle interactions (Saminathan et al, 2022), which play a key role in the regulation of biological processes. Thus, our finding shed a light to the novel field regarding the relationship between organelle interactions and HSC regulations.

## Methods

### Mice

Mitol$^{Flox/Flox}$ mice were crossed with MX1-cre +/− mice to obtain Mitol$^{Flox/Flox}$; MX1-cre +/− mice. These mice were intraperitoneally injected with polyinosinic-polycytidylic acid (Poly:IC, Invivogen, 200 mg/kg body weight, thrice every other day) to induce cre-mediated recombination and were considered as Mitol$^{\Delta/\Delta}$ following PCR confirmation of Mitol gene deletion. Poly:IC injection schedule was adjusted to analyze the mice at around 8–12 weeks of age. Mitol$^{Flox/Flox}$; MX1-cre −/− littermates were similarly injected with Poly:IC and were considered as Mitol$^{Flox/Flox}$ control mice. All experiments related to mice were performed according to the guidelines of Kumamoto University animal use.

### Bone marrow and peripheral blood analysis

Total bone marrow (BM) cells were obtained by flushing femur and tibia bones with Dulbecco's Modified Eagle Medium (DMEM, Sigma) containing 10% fetal bovine serum (FBS, Biowest). Nucleated cells were counted after treating the BM cells with Turk's solution. Peripheral blood was collected from the eyeball and analyzed by automatic hematology analyzer. To identify different HSPC population, BM cells were stained with fluorescence-conjugated antibodies against different cell surface markers and analyzed by flow cytometry. Following antibodies were used: c-Kit (2B8, PE-Cy7; BD Biosciences), Sca1 (E13-161.7, FITC; eBioscience), CD150 (TC15-12F12.2, BV-421; BioLegend), CD48 (HM48-1, APC-Cy7; BioLegend), EPCR (eBio1560, APC; eBioscience), Ter-119 (TER-119, PerCP-Cy5.5; BD Biosciences), CD4 (GK1.5, PerCP-Cy5.5; BD Biosciences), CD8a (53-6.7, PerCP-Cy5.5; BD Biosciences), B220 (RA3-6B2, PerCP-Cy5.5; BD Biosciences), Gr-1 (RB6-8C5, PerCP-Cy5.5; BD Biosciences), Mac-1 (M1/70, PerCP-Cy5.5; BD Biosciences). Cell population was identified as the following definition: HSPCs (Lineage- c-Kit+ EPCR + , LEK), MPPs (LEK, CD48 + ), ST-HSC (LEK, CD48-CD150-), and LT-HSC (LEK, CD48− CD150 + ). Flow cytometric analysis was performed with BD FACSCanto™ II and Flowjo software ver.10 was used for data analysis.

### Transplantation

Transplantation experiments were performed following the previously described method (Hashimoto et al, 2021). Briefly, C57BL/6 Ly5.1 mice were used as recipients. These mice were lethally irradiated and the following day transplanted with $1 \times 10^6$ test BM cells (Ly5.2) along with $1 \times 10^6$ competitor cells (Ly5.1). Reconstitution of PB and BM population with transplanted cells were analyzed by flow cytometry at different time points as mentioned in the results section. To analyze chimerism maintenance potential, total BM cells were transplanted without prior Poly:IC induction. Six weeks post transplantation the mice were injected with Poly:IC to induce Cre-mediated knockout of Mitol gene. These mice were similarly analyzed for PB and BM reconstitution using flow cytometry.

### Mitochondrial metabolism analysis

Mitochondrial membrane potential (MMP) and mitochondrial superoxide production of HSPCs were determined respectively with Mitoprobe JC-1 assay kit (Thermo Fisher Scientific) and MitoSOX Red Mitochondrial Superoxide Indicator (Thermo Fisher Scientific), following the manufacturer's instructions.

Mitochondrial respiration of LSK cells was performed in Seahorse XF HS Mini Analyzer (Agilent Tech.) following the manufacturer's protocol with necessary optimization. For final analysis, 50,000 LSK cells were seeded in each test well and Mito Stress Test was run with 1.5 μM Oligomycin, 5.0 μM FCCP, and 1.0 μM Rot/AA. For LSK cell sorting, firstly c-Kit$^+$ cells were enriched from whole BM cells by using autoMACS Pro Separator (Militenyl Biotec) following the treatment with magnetic beads conjugated anti-c-kit antibody (Militenyl Biotec). After c-Kit$^+$ cell-enriched fraction was then stained with corresponding antibodies for Sca1, c-Kit, and Lineage markers as described above, LSK cells (Lineage$^-$ Sca1$^+$ c-Kit$^+$) were sorted by using MoFlo XDP Cell Sorter (Beckman Coulter).

### Intracellular marker staining

To detect intracellular cell markers related to ER stress pathway, cells were treated with BD Cytofix-Cytoperm solution to permeabilize the cell membrane. After that, cells were incubated with antibodies against XBP-1s (CST) and pEIF2a (CST) diluted in 1× BD Wash buffer, and analyzed in FACS Canto II. For cell cycle analysis, similar procedure was followed for staining with anti-Ki67 antibody, with additional Hoechst33342 incubation as the final step before flow cytometry.

### Cell culture and apoptotic cell detection

To assess cellular apoptosis, LT-HSCs were sorted and cultured for 36 hours before staining with Annexin V - APC probe (BD

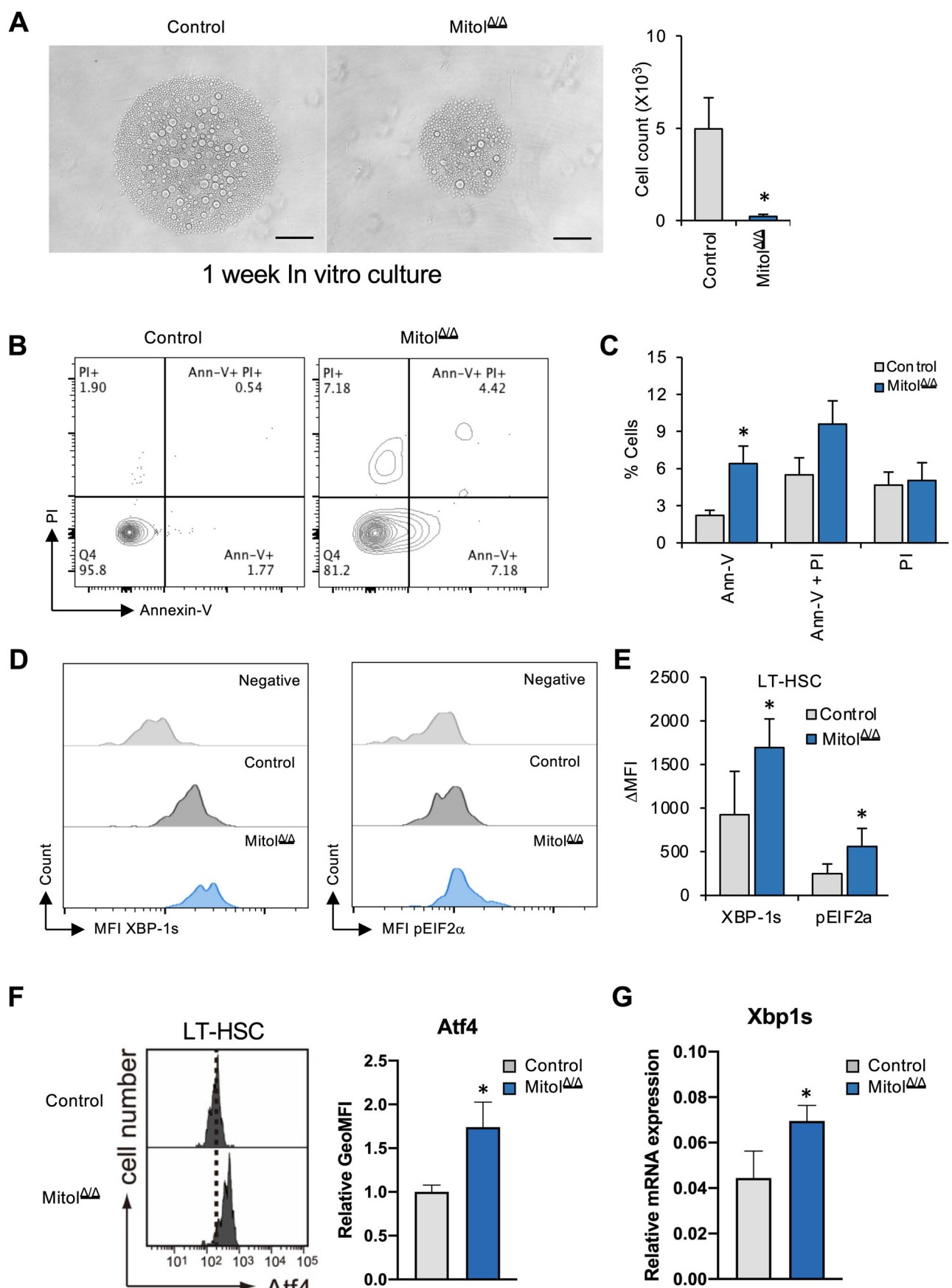

◀ **Figure 5. Mitol deletion induces ER stress-mediated apoptosis in HSCs.**

(A) Detectable viable cell counts after 1 week of in vitro culture of 100 LT-HSCs ($n = 6$). (B, C) Cellular apoptosis of LT-HSCs shown by FACS (B) following 36 h of in vitro culture, apoptotic cell count is shown in (C) ($n = 10$). (D, E) FACS analysis showing the levels of two ER pathway markers (XBP-1s and pEIF2a) in HSCs. A representative experiment of $n = 4$. (F) FACS analysis showing relative expression levels of Aft4 in HSCs ($n = 3$). (G) qRT-PCR analysis showing relative mRNA expression levels of Xbp-1s in HSCs ($n = 3$). Data Information: (A) the scale bar is 200 µm. Data represent mean ± SEM with two-tailed unpaired Student's $t$ test. ns, $P > 0.05$; *$P < 0.05$. Source data are available online for this figure.

Pharmingen). For LT-HSC sorting, total bone marrow cells were first enriched with anti-c-kit MACS beads using autoMACS Pro Separator (Militenyl Biotec). This c-Kit enriched cell fraction was then stained with HSC surface markers as mentioned earlier and LT-HSCs were sorted using MoFlo XDP Cell Sorter (Beckman Coulter). Cells were directly sorted into 96-well U-bottomed plates containing S-Clone SF-03 culture medium (EIDIA) supplemented with 10 ng/ml mouse stem cell factor (R&D systems, Minneapolis, MN) and 100 ng/ml mouse thrombopoietin (R&D systems) and 0.5% fatty acid -free bovine serum albumin (BSA) (Proliant Biologicals, Ankeny, IA). Annexin V staining (BD Pharmingen) was done following the protocol provided by the manufacturer.

## ER stress inhibitor

Murine HSCs were collected and cultured following the methods described above in Cell culture section. Briefly, 1000/well LT-HSCs were sorted into 96-well U-bottomed plate, containing S-Clone SF-03 culture medium (EIDIA) supplemented with 10 ng/ml mouse stem cell factor (SCF) (R&D systems, Minneapolis, MN) and 100 ng/ml mouse thrombopoietin (TPO) (R&D systems) and 0.5% fatty acid -free bovine serum albumin (BSA) (Proliant Biologicals, Ankeny, IA). Subsequently, KIRA6 (Selleckchem; S8658), an ER stress inhibitor, were added into cultured LT-HSCs with final concentrations of 0.03 µM and 0.1 µM, respectively. After incubation at 37 °C for 20 h, cells were collected and stained with PI and Annexin V - APC probe (BD Pharmingen) following the protocol provided by the manufacturer. Apoptotic populations were detected and analyzed by flow cytometry Aria IIIu (BD Sciences).

## RNA-seq

As described previously (Umemoto et al, 2018), 100 sorted HSCs were lysed followed by first-strand cDNA synthesis using Prime-Script RT reagent kit (Takara) and not-so-random primers. Subsequently, the second strand was synthesized using Klenow Fragment (3'–5' exo-; New England Biolads). After purification of the double-stranded cDNA, the library for RNA-seq was prepared by a Nextera XT DNA sample Prep Kit (Illumina), according to the manufacturer's instruction. The prepared library was sequenced on a Next-Seq system (Illumina) at 75 bp of single-end read.

## Data analysis of RNA-seq

The FASTQ files containing raw bulk RNA-seq reads were aligned to mouse reference genome (mm39) using Spliced Transcripts Alignment (STAR) pipeline (Dobin et al, 2013), the generated BAM files were then merged using samtools and transcripts were counted using subread package. The output count matrix was imported into R package DESeq2 for downstream analysis. DESeq2 object was created using DESeqDataSetFromMatrix function of DESeq2

package, and differential gene expression analysis based on the negative binomial distribution was performed using DESeq function, with wild-type group set as the reference. Next, the output list containing differentially expressed genes (DEGs) was used for biological pathway analysis. Software information is shown in Table EV2.

## Differential gene expression analysis

A DESeq2 object was generated using the DESeqDataSetFromMatrix function from the DESeq2 package (Love et al, 2014). DESeq2 employs size factor normalization to address variations in sequencing depth among samples and correct for library-specific biases. The size factors are computed for each sample, based on the median ratio of observed read counts to the geometric mean of gene-specific counts. Subsequently, the raw counts for each gene in every sample are adjusted by dividing them by their corresponding size factors (i.e., library size adjustment), and the adjusted counts are then log-transformed. Following the normalization step, differential gene expression analysis using the negative binomial distribution by the Wald test was conducted using the DESeq function, with the wild-type group designated as the reference. The resulting list of differentially expressed genes (DEGs) was further utilized for biological pathway analysis.

## Unsupervised biological pathway analysis

The genes from the DEGs list were ranked based on their difference size (DS) score, defined as their "log$_2$FoldChange" multiplied by "minus log$_{10}$Adjusted $P$ value". Next, top-500 upregulated and top-500 downregulated genes (with positive and negative DS, respectively) were imported into Enrichr, a web-based tool (Xie et al, 2021), to identify the enriched biological pathways, in an unsupervised manner. Finally, a curated list of enriched pathways was visualized as previously described (Bonnot et al, 2019).

## Pathway gene set score quantification

The gene list for each Gene Ontology (GO) term of interest was obtained from Mouse Genome Informatics (MGI), their average expression (gene set score) was calculated using normalized count values in each sample, and statistically compared across the two experimental conditions by Welch two-sample $t$ test. A difference with $P$ value less than 0.05 was considered as statistically significant.

## Blood cell analysis

Peripheral blood was collected by retro-orbital bleeding under anesthesia, and complete blood cell counts were performed using an automated hematology analyzer, pocH-100i V Diff (Sysmex).

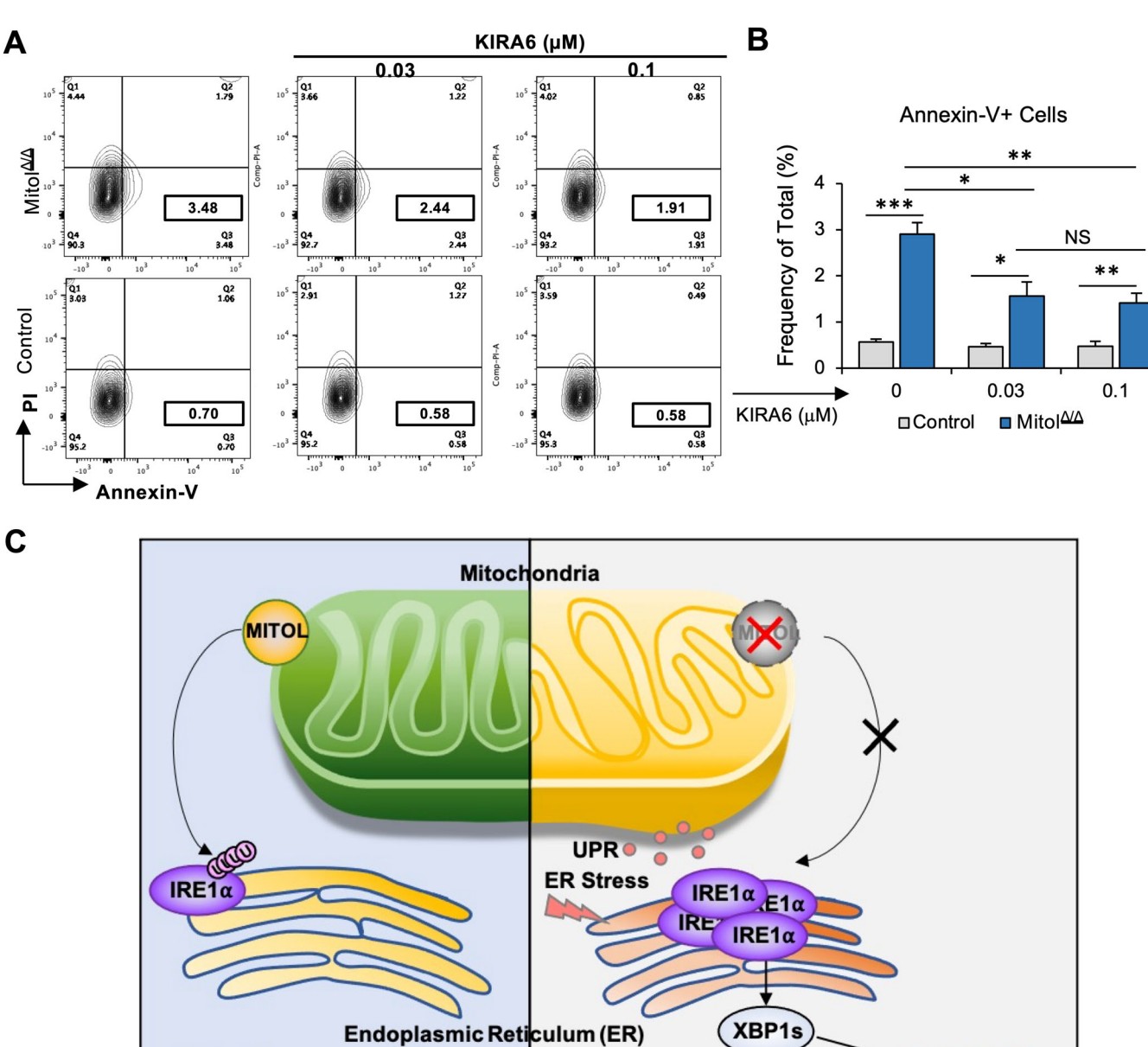

**Figure 6.  IRE1α inhibitor partially protects HSCs from ER stress-induced apoptosis caused by Mitol deletion.**

(A, B) Cellular apoptosis analysis of LT-HSCs 1 week post Poly:IC induction and treated with different dose of KIRA6 shown by FACS plot (A) and a graph shown in (B) (*n* = 4). (C) Schematic figure of MITOL functions in HSCs. MITOL is essential to maintain HSCs homeostasis through ubiquitin-mediated degradation of IRE1α. Mitol deletion is shown to initiate ER stress, which triggers cellular apoptosis that is mainly regulated by IRE1α signaling. Data Information: Data represent mean ± SD with two-tailed unpaired Student's *t* test. ns, *P* > 0.05; *P* < 0.05; **P* < 0.01; ***P* < 0.001. Source data are available online for this figure.

## Measurement of glucose uptake

According to the manufacturer's instructions of Glucose Uptake Assay Kit-Green (DOJINDO LABORATORIES, Kumamoto, Japan), c-Kit⁺ cells enriched by AutoMACS pro (Miltenyi Biotec, North Rhine-Westphalia, Germany) were cultured with Glucose Uptake Probe-Green at 37 °C for 30 min following pre-culture at 37 °C for 15 min. After staining for the identification of LT-HSCs, the potential for glucose uptake was assessed based on the fluorescence intensity of glucose probe using a flow cytometer.

## Flow cytometric analyses for ATF4

Prior to staining with anti-ATF4 antibody (Novus Biologicals, Centennial, CO), we labeled the antibody by using Zenon Rabbit IgG Labeling Kits (Thermo Fisher Scientific Inc., Waltham, MA). To cell fixation and permeabilization, we used a PerFix

EXPOSE kit (Beckman Coulter) according to the manufacturer's instructions.

## RNA extraction and quantitative PCR

Total RNA from LT-HSCs was extracted using ISOGEN (Wako). Reverse transcription was performed using a PrimeScript RT reagent Kit (TaKaRa) following the manufacturer's instruction. Gene expression was determined using Luna Universal qPCR Master Mix (BioLabs) using Roche LightCycler96 system. Data was normalized relative to *Rps18*. The primer sequences used are listed in Table EV1. To detect spliced Xbp1 (Xbp-1s), we employed RamDA-RT with minor modifications as described previously (Umemoto et al, 2018) and Xbp-1s-specific TaqMan® MGB probe (Assay ID: Mm03464496_m1; Thermo Fisher Scientific Inc., Waltham, MA).

## Statistical analysis

Results are expressed as mean ± SEM/SD. Three-group comparisons were analyzed by one-way ANOVA and Holm–Sidak's multiple-comparisons test. Two-group comparisons were carried out with the Welch's two-sided *t* test. Statistical analyses were performed with Prism 8 (GraphPad). Significance levels of *P* values are shown as: ns, $P > 0.05$; *$P < 0.05$; **$P < 0.01$; ***$P < 0.001$; ****$P < 0.0001$.

## Data availability

RNA-seq data is available to the public (GSE240788). See GEO accession GSE240788 for review.

## Peer review information

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

## Acknowledgements

We thank Miho Kataoka for mouse maintenance and technique assistance and Alban Johansson for technique assistance. T. Suda is supported by the Singapore Translational Research Investigator Award from the National Medical Research Council in Singapore (NMRC/STaR 18 may-0004) and in part by a JSPS KAKENHI Grant-in-Aid for Scientific Research (S; grant no. 26221309).

## Author contributions

**Wenjuan Ma**: Conceptualization; Resources; Data curation; Software; Formal analysis; Validation; Investigation; Methodology; Writing—original draft; Writing—review and editing. **Shah Adil Ishtiyaq Ahmad**: Conceptualization; Resources; Data curation; Software; Formal analysis; Validation; Investigation; Methodology; Writing—original draft; Writing—review and editing. **Michihiro Hashimoto**: Conceptualization; Resources; Data curation; Software; Formal analysis; Validation; Investigation; Methodology. **Ahad Khalilnezhad**: Resources; Data curation; Software; Formal analysis; Validation; Methodology. **Miho Kataoka**: Validation; Investigation; Methodology. **Yuichiro Arima**: Investigation; Methodology. **Yosuke Tanaka**: Methodology; Writing—review and editing. **Shigeru Yanagi**: Resources. **Terumasa Umemoto**: Conceptualization; Resources; Data curation; Software; Formal analysis; Supervision; Validation; Investigation; Methodology; Project administration; Writing—review and editing. **Toshio Suda**: Conceptualization; Resources; Data curation; Supervision; Funding acquisition; Investigation; Project administration; Writing—review and editing.

## Disclosure and competing interests statement

The authors declare no competing interests. T.S. is a member of the *EMBO Journal* editorial advisory board.

# Expanded View Figures

**Figure EV1. Features of Mitol$^{\Delta/\Delta}$ HSCs.**

(A) Peripheral blood populations after 1 weeks of Poly:IC induced Mitol deletion ($n = 3$). (B) PCA analysis of RNA-seq generated from murine control and Mitol$^{\Delta/\Delta}$ HSCs after 1 week post Poly:IC induction. (C) Heatmap of top 100 DEGs from RNA-seq analysis. (D, E) Cell cycle analysis of HSPCs 1 week post Poly:IC induction shown by FACS (D) and cell percentage at different cell cycle stages shown in (E) ($n = 3$). Data Information: Data represent mean ± SEM with two-tailed unpaired Student's *t* test. ns, $P > 0.05$; *$P < 0.05$; **$P < 0.01$.

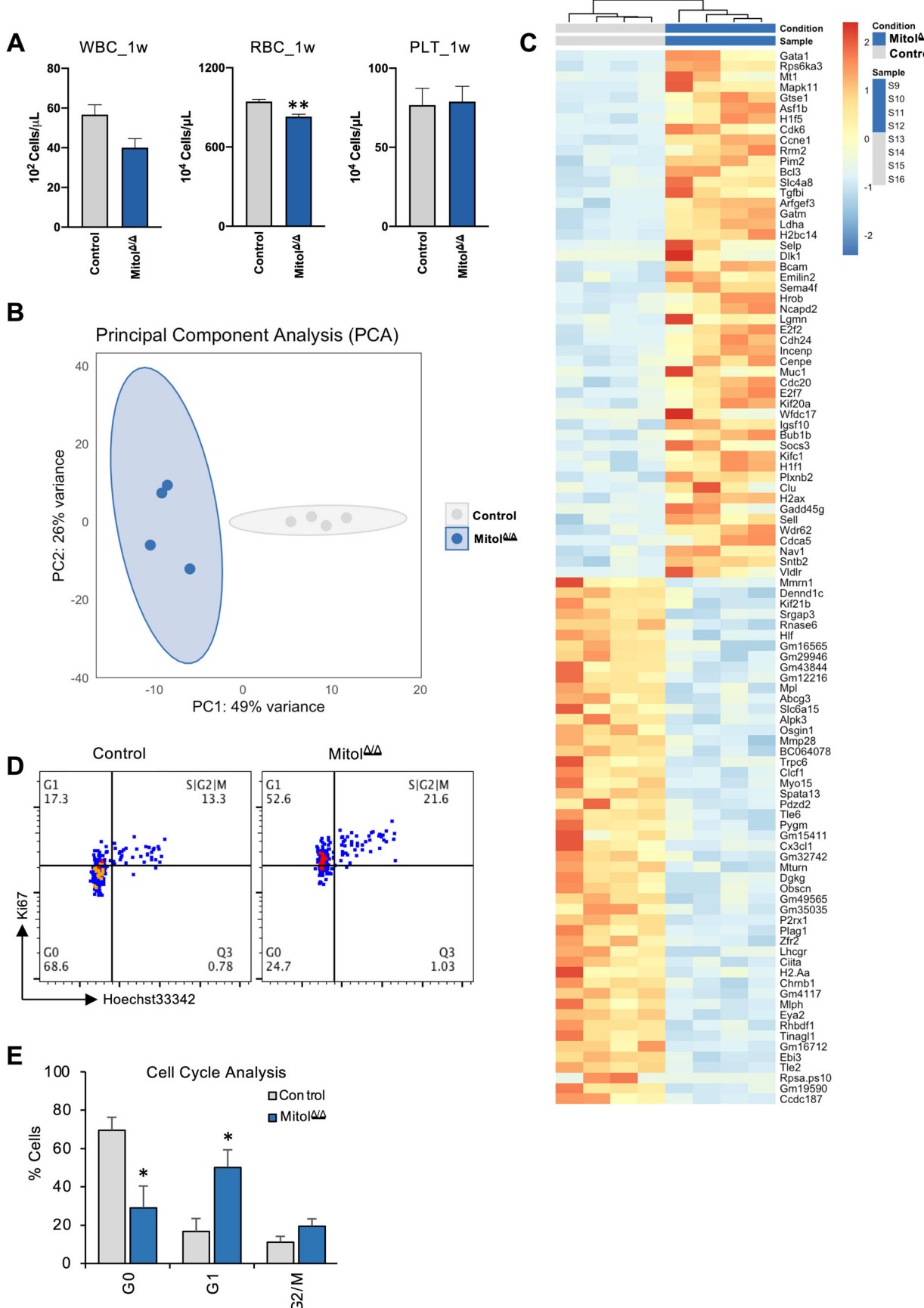

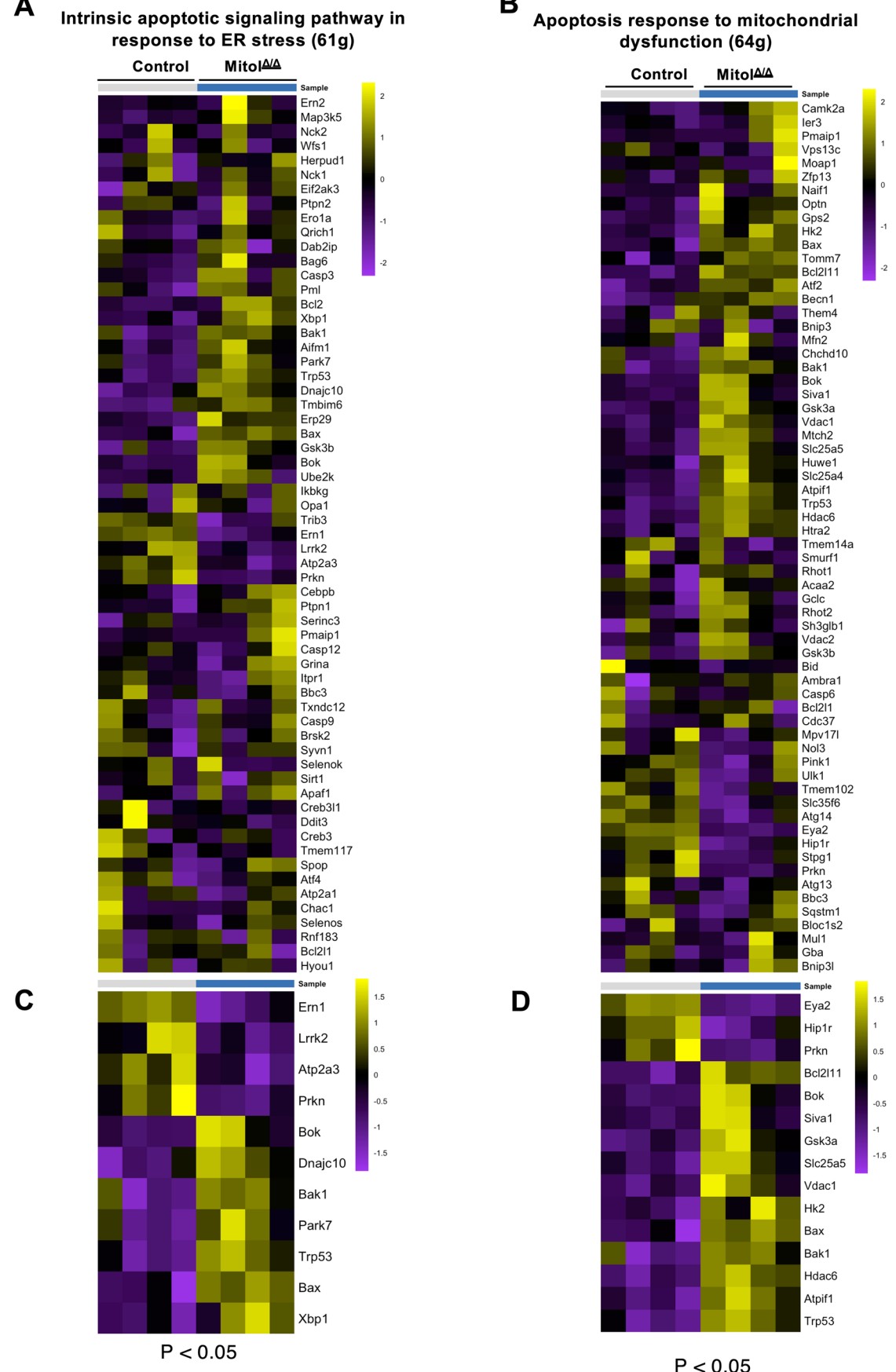

**A** Intrinsic apoptotic signaling pathway in response to ER stress (61g)

**B** Apoptosis response to mitochondrial dysfunction (64g)

**C** P < 0.05

**D** P < 0.05

**Figure EV2.  Genes involved in ER and mitochondrial-regulated apoptosis.**

(**A–D**) Heatmaps of Intrinsic apoptotic signaling pathway in response to ER stress (**A**) with significantly upregulated genes (**C**) and apoptosis response to mitochondrial dysfunction (**B**) with significantly upregulated genes (**D**). Data Information: Data was analyzed with the Wald test.

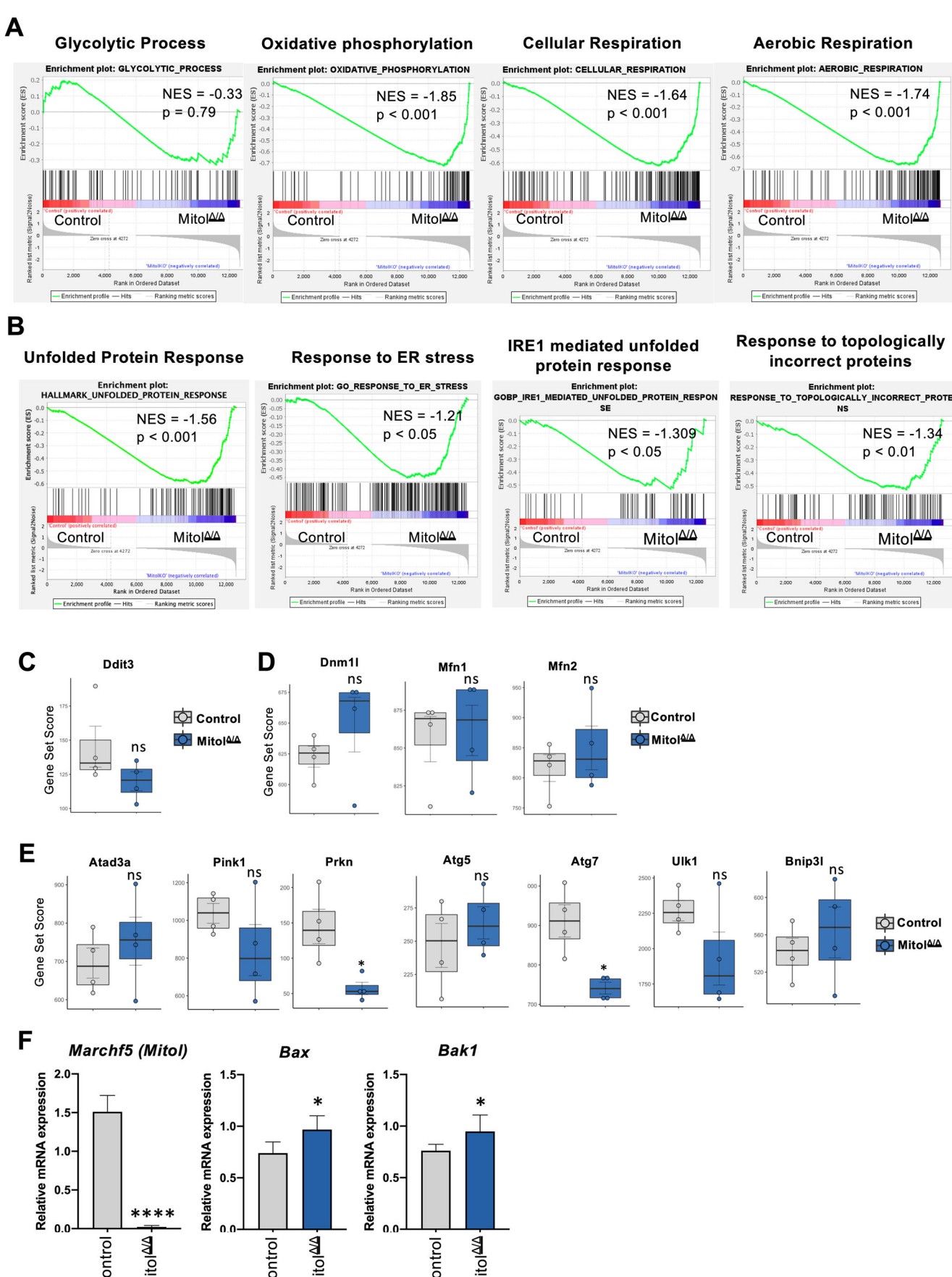

◀

**Figure EV3.  Mitol involvement in mitochondrial function and mitophagy.**

(**A, B**) GSEA analysis of metabolic genes involved in ER and mitochondrial activity. (**C**) Gene encoding CHOP ($n = 4$). (**D**) Genes encoding DRP1, MFN1 and MFN2, respectively, which are involved in mitochondrial fusion and fission ($n = 4$). (**E**) Genes which are involved in Pink-Parkin mediated mitophagy pathway ($n = 3$). (**F**) RT-qPCR analysis showing genes expression levels in murine LT-HSC after 1 week of Poly:IC injection ($n = 5$). Data Information: In EV3A, B, $P$ value of the normalized enrichment score (NES) was analyzed with an empirical phenotype-based permutation test. In EV3C-E, the box plot's central band marks the median, boxes mark the first and third quartiles, and whiskers extend the boxes to the largest value no further than 1.5 times the interquartile range. Gene Set Score was analyzed with the Wald test. In EV3F, data represent mean ± SEM with two-tailed unpaired Student's $t$ test. ns, $P > 0.05$; $*P < 0.05$; $****P < 0.0001$.

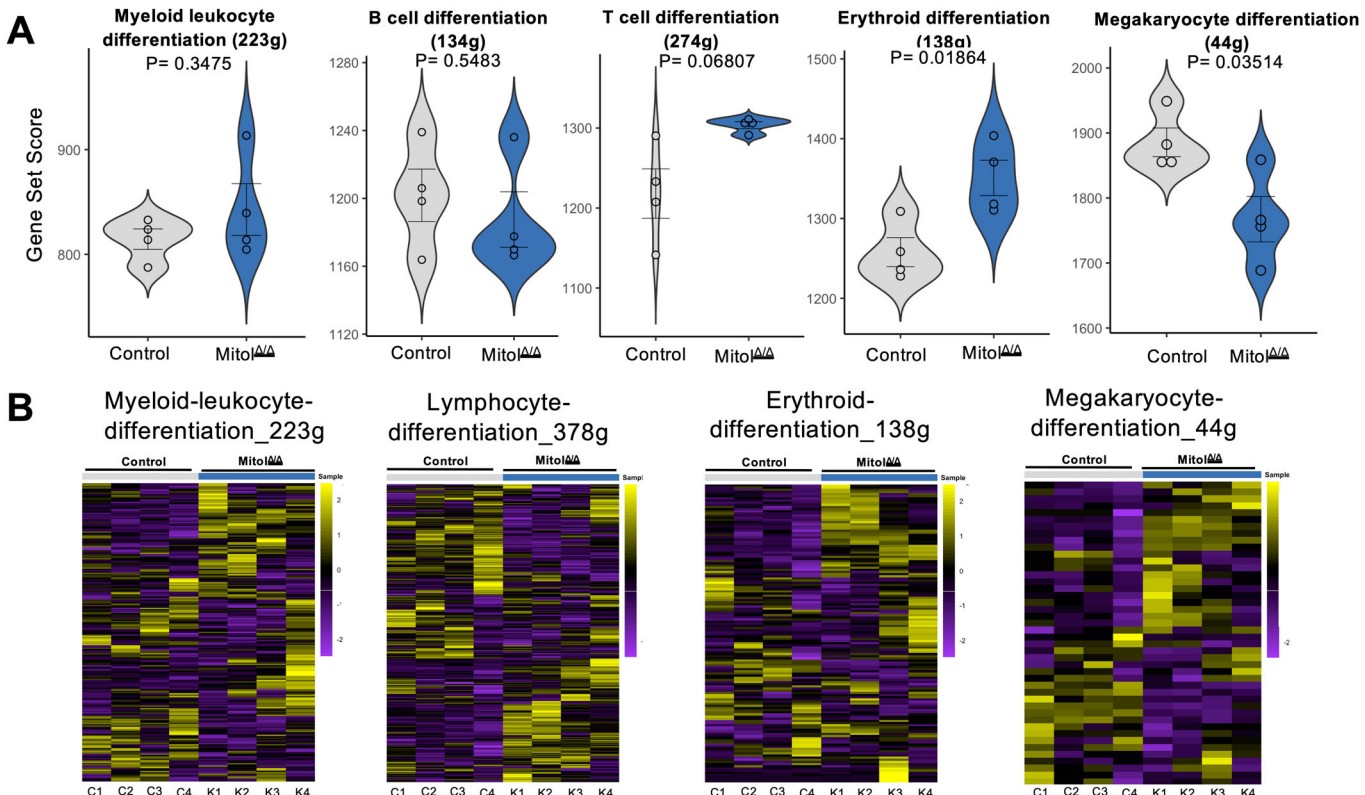

**Figure EV4. Distribution of lineage programming of Mitol^ΔΔ HSCs shown in RNA-seq analysis.**

(A, B) Distribution of lineage programs in Mitol^ΔΔ HSCs 1 week post Poly:IC induction was shown in (A) Violin plots and heatmaps (B) by RNA-seq analysis (*n* = 4). Data Information: In EV4A, horizontal lines in violin plots represent quartiles. Data was analyzed with the Wald test.

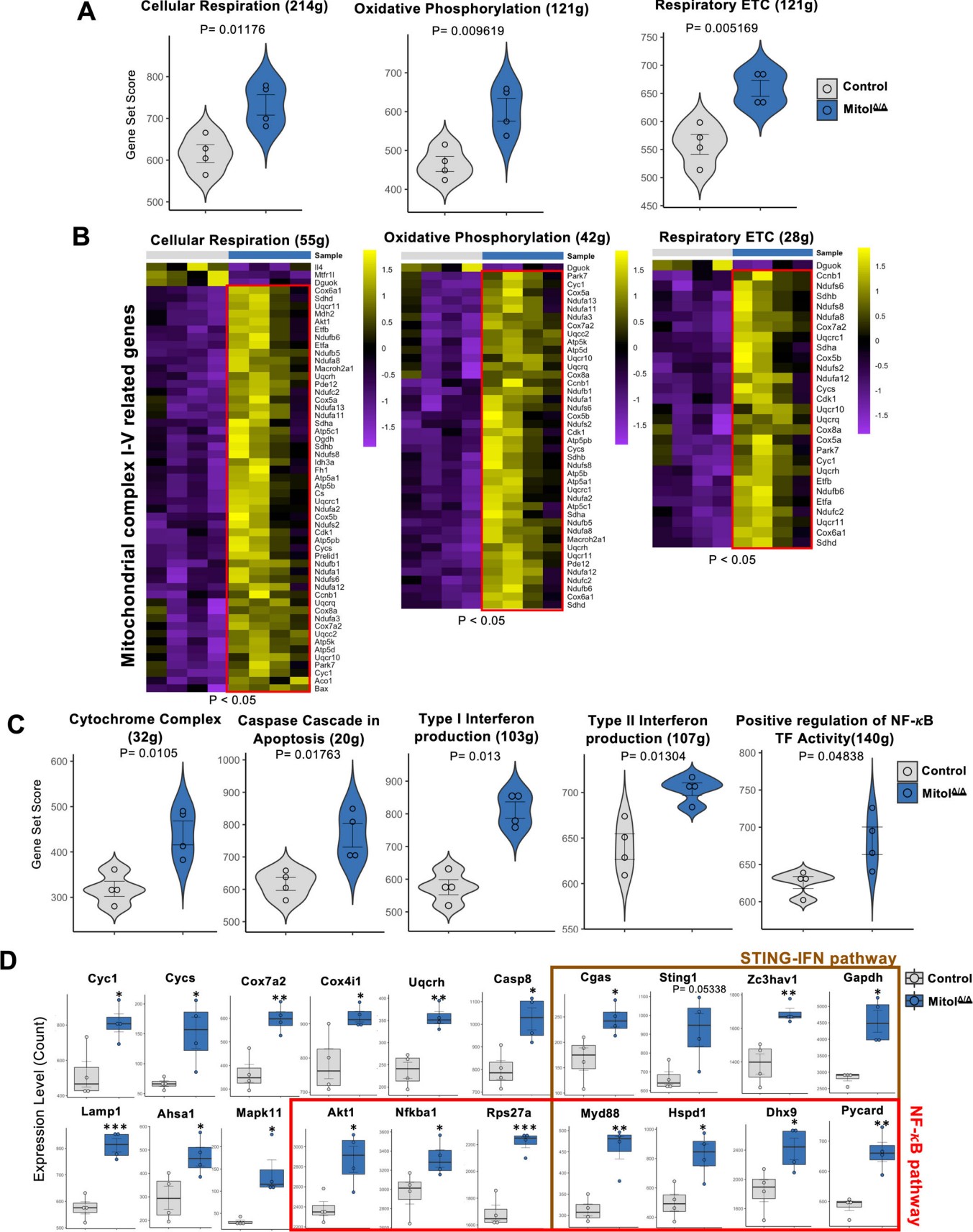

◀ **Figure EV5. Enhanced mitochondrial respiration by Mitol regulates apoptotic pathways.**

(A) Violin plots showed enhanced mitochondrial respiration in Mitol$^{\Delta/\Delta}$ HSCs ($n = 4$). (B) Heatmaps showed significantly upregulated genes in mitochondrial respiration caused by Mitol depletion. Red box highlights genes encoding mitochondria complex I–V. (C) Violin plots showed activated pathways associated with enhanced mitochondrial metabolism in Mitol$^{\Delta/\Delta}$ HSCs ($n = 4$). (D) Box plots showed significantly upregulated genes involved in pathways shown in violin plots (C) ($n = 4$). Borrow color highlights genes involved in STING-IFN pathway and red color highlights genes involved in NF-κB pathway. Data Information: In EV5A,C, horizontal lines in violin plots represent quartiles. In EV5D, the box plot's central band marks the median, boxes mark the first and third quartiles, and whiskers extend the boxes to the largest value no further than 1.5 times the interquartile range. Data was analyzed with the Wald test. ns, $P > 0.05$; *$P < 0.05$; **$P < 0.01$; ***$P < 0.001$.

