## [Peer Review File · The EMBO Journal]

Mitol deficiency triggers hematopoietic stem cell apoptosis via ER stress response

Wenjuan Ma, Shah Ahmad, Michihiro Hashimoto, Ahad Khalilnezhad, Miho Kataoka, Yuuichiro Arima, Yosuke Tanaka, Shigeru Yanagi, Terumasa Umemoto, and Toshio Suda

DOI: [10.15252/emboj.2023114405](https://doi.org/10.15252/emboj.2023114405)

Corresponding authors: Toshio Suda (csits@nus.edu.sg) , Terumasa Umemoto (umemoto@kumamoto-u.ac.jp)

Review Timeline:

Submission Date:	29th Apr 23
Editorial Decision:	23rd Jun 23
Revision Received:	19th Nov 23
Editorial Decision:	12th Dec 23
Revision Received:	18th Dec 23
Accepted:	20th Dec 23

Editor: Daniel Klimmeck

Transaction Report:

Dear Toshio,

Thank you again for the submission of your manuscript (EMBOJ-2023-114405) to The EMBO Journal, as well as for your patience with our response at this time of the year which got protracted due to delayed referee input and discussions in the editorial team. Your study has been sent to two reviewers with expertise in hematopoietic stem cell biology and bioenergetic stress signaling and we have received feedback from both of them, which I enclose below.

As you will see, the referees acknowledge the potential interest and novelty of your comparative analysis, although they also express several issues that will have to be conclusively addressed before they can be supportive of publication of your manuscript in The EMBO Journal. We judge the comments of the referees to be generally reasonable and given their overall interest, we are happy to invite you to revise your manuscript experimentally to address the referees' comments.

As you may remember from previous exchange, we generally allow three months as standard revision time. As a matter of policy, competing manuscripts published during this period will not negatively impact on our assessment of the conceptual advance presented by your study. However, we request that you contact the editor as soon as possible upon publication of any related work, to discuss how to proceed. Should you foresee a problem in meeting this three-month deadline, please let us know in advance and we may be able to grant an extension.

When submitting your revised manuscript, please carefully review the instructions below.

Please feel free to approach me any time should you have any questions related to this.

Thank you for the opportunity to consider your work for publication.

I look forward to your revision.

Best regards,

Daniel

Daniel Klimmeck, PhD
Senior Editor
The EMBO Journal

Instruction for the preparation of your revised manuscript:

- 1) a .docx formatted version of the manuscript text (including legends for main figures, EV figures and tables). Please make sure that the changes are highlighted to be clearly visible.
- 2) individual production quality figure files as .eps, .tif, .jpg (one file per figure).
- 3) a .docx formatted letter INCLUDING the reviewers' reports and your detailed point-by-point response to their comments. As part of the EMBO Press transparent editorial process, the point-by-point response is part of the Review Process File (RPF), which will be published alongside your paper.
- 4) a complete author checklist, which you can download from our author guidelines ([https://wol-prod-cdn.literatumonline.com/pb-assets/embo-site/Author Checklist%20-%20EMBO%20J-1561436015657.xlsx](https://wol-prod-cdn.literatumonline.com/pb-assets/embo-site/Author%20Checklist%20-%20EMBO%20J-1561436015657.xlsx)). Please insert information in the checklist that is also reflected in the manuscript. The completed author checklist will also be part of the RPF.
- 5) Please note that all corresponding authors are required to supply an ORCID ID for their name upon submission of a revised manuscript.
- 6) It is mandatory to include a 'Data Availability' section after the Materials and Methods. Before submitting your revision, primary datasets produced in this study need to be deposited in an appropriate public database, and the accession numbers and database listed under 'Data Availability'. Please remember to provide a reviewer password if the datasets are not yet public (see <https://www.embopress.org/page/journal/14602075/authorguide#datadeposition>). In case you have no data that requires deposition in a public database, please state so in this section. Note that the Data Availability Section is restricted to new primary data that are part of this study.

7) Our journal encourages inclusion of *data citations in the reference list* to directly cite datasets that were re-used and obtained from public databases. Data citations in the article text are distinct from normal bibliographical citations and should directly link to the database records from which the data can be accessed. In the main text, data citations are formatted as follows: "Data ref: Smith et al, 2001" or "Data ref: NCBI Sequence Read Archive PRJNA342805, 2017". In the Reference list, data citations must be labeled with "[DATASET]". A data reference must provide the database name, accession number/identifiers and a resolvable link to the landing page from which the data can be accessed at the end of the reference. Further instructions are available at .

8) At EMBO Press we ask authors to provide source data for the main and EV figures. Our source data coordinator will contact you to discuss which figure panels we would need source data for and will also provide you with helpful tips on how to upload and organize the files.

Numerical data can be provided as individual .xls or .csv files (including a tab describing the data). For 'blots' or microscopy, uncropped images should be submitted (using a zip archive or a single pdf per main figure if multiple images need to be supplied for one panel). Additional information on source data and instruction on how to label the files are available at .

9) We replaced Supplementary Information with Expanded View (EV) Figures and Tables that are collapsible/expandable online (see examples in <https://www.embopress.org/doi/10.15252/emj.201695874>). A maximum of 5 EV Figures can be typeset. EV Figures should be cited as 'Figure EV1, Figure EV2' etc. in the text and their respective legends should be included in the main text after the legends of regular figures.

10) When assembling figures, please refer to our figure preparation guideline in order to ensure proper formatting and readability in print as well as on screen:
<http://bit.ly/EMBOPressFigurePreparationGuideline>

11) For data quantification: please specify the name of the statistical test used to generate error bars and P values, the number (n) of independent experiments (specify technical or biological replicates) underlying each data point and the test used to calculate p-values in each figure legend. The figure legends should contain a basic description of n, P and the test applied. Graphs must include a description of the bars and the error bars (s.d., s.e.m.).

We realize that it is difficult to revise to a specific deadline. In the interest of protecting the conceptual advance provided by the work, we recommend a revision within 3 months (21st Sep 2023). Please discuss the revision progress ahead of this time with the editor if you require more time to complete the revisions.

Referee #1:

In this interesting manuscript, Ma W et al. describe the role of the mitochondrial membrane-localized E3 ubiquitin ligase Mitol/Marchf5 (encoded by Mitol gene) in HSCs. Mitol is known to regulate mitochondrial and endoplasmic reticulum (ER) interaction and promote cell survival. The authors demonstrate that Mitol is required to protect HSCs from apoptosis, through the protective branch of ER. Overall, the data are compelling and the manuscript is very well written. Below are some points that require clarification:

1. To make the transcript count comparable, the normalisation of raw counts is needed. I believe it has been done in the RNA-seq data processing. However, could authors describe the normalisation method for transcript counts in the method? It would be recommended to highlight the 'normalised counts' in Fig 3F and 3I.
2. For multiple gene comparisons in RNAseq data, p value is required to be corrected to reduce the false positive rate (FDR). However, raw P value is used in Fig 3A. Likewise, the two-sample t-test and raw P value are used in Fig 3F and 3I. A more stringent analysis would be recommended.
3. Why did authors prefer gene set score in this paper (average expression level of genes in specific gene-set)? This is not the best or most common way to evaluate pathway activity in this type of study. Over Representation Analysis (ORA), Gene Set Enrichment Analysis (GSEA) and Gene Set Variation Analysis (GSVA) may be better suited and less biased options.
4. Three pathways in unfolded protein response are initiated simultaneously and display dissimilar dynamics under the proteotoxic stress. As XBP1s and p-EIF2a are both observed in Mitol^{-/-} HSC, why do the authors interpret that IRE1a-XBP1 pathway is the major contributor to the protective mechanism? While unfolded protein response or endoplasmic reticulum stress are clear mediators, the role of IRE1a is more tenuous. To further support this UPR/ER stress hypothesis, ATF4 protein may be a good and robust marker to measure UPR activity. If possible, Xbp1 splicing assay by qRT-PCR is recommended as an alternative method to support the IRE1a-XBP1 pathway.
5. Seahorse assay is not really "In vivo", but "ex vivo" measurement of mitochondrial respiration. It will be helpful if the authors could explain how LSK cells are collected for Seahorse assay in the Methods.

Referee #2:

Ma and colleagues provide a novel insight into the function of the outer mitochondrial membrane-localized E3 ubiquitin ligase Mitol/MARCH5 in hematopoietic stem cells (HSCs). They revealed that Mitol plays an important role in HSC function and quiescence. They nicely showed that Mitol deficient HSC have prolonged ER stress which induces IRE1a signaling. Blockage of ER stress pharmacologically with KIRA6 rescues the HSC defects by reducing apoptosis.

Overall, this is a very well-written and presented manuscript describing an interesting and mechanism regulating stem cell homeostasis. The authors present their findings clearly. A real strength of this study is the identification of a novel mechanism of HSCs quiescence regulation. These findings are clearly of high relevance in the context of better understanding how to overcome stress induced apoptosis in HSC. However, there are some open questions in the study design and the current set of data as listed below.

Below are comments that aim to further increase the clarity and significance of this exciting study:

1. Can the authors please provide peripheral blood analysis from 1 week post-Poly:IC mice?
2. Can the authors comment whether Mitol deficient HSC in Fig.2A are able to home to the bone marrow?
3. In Fig.4C the authors measured MitoSox levels in Mitol deficient mice, however the histograms show a bimodal distribution of the dye. The MitoSox high peak is even smaller in the knockout mice. Can the authors comment on this finding?
4. The authors should provide more insights in difference in glucose uptake by using 2-NBDG in wildtype vs. Mitol-deficient mice.
5. The authors should also provide Seahorse data from 4 week post-Poly:IC mice to better understand the metabolic adaptations of Mitol deficient LSK cells.
6. The authors nicely show that IRE1a inhibition with KIRA6 rescues the phenotype in vitro. The authors should show the same effect in vivo treating mice with KIRA6 (PMID: 25018104).

Ma W et al: Response to Reviewers' comments:**Referee #1:**

In this interesting manuscript, Ma W et al. describe the role of the mitochondrial membrane-localized E3 ubiquitin ligase Mitol/Marchf5 (encoded by Mitol gene) in HSCs. Mitol is known to regulate mitochondrial and endoplasmic reticulum (ER) interaction and promote cell survival. The authors demonstrate that Mitol is required to protect HSCs from apoptosis, through the protective branch of ER. Overall, the data are compelling and the manuscript is very well written. Below are some points that require clarification:

1. To make the transcript count comparable, the normalization of raw counts is needed. I believe it has been done in the RNA-seq data processing. However, could authors describe the normalisation method for transcript counts in the method? It would be recommended to highlight the 'normalised counts' in Fig 3F and 3I.

Reply:

We thank the reviewer for the constructive comment. The normalization method for transcript counts is added in the Materials and methods (Page 19; line 462-473) and the description is shown as following,

Differential Gene Expression Analysis

A DESeq2 object was generated using the DESeqDataSetFromMatrix function from the DESeq2 package (Love et al., *Genome Biol*, 2014; 15: 550). DESeq2 employs size factor normalization to address variations in sequencing depth among samples and correct for library-specific biases. The size factors are computed for each sample, based on the median ratio of observed read counts to the geometric mean of gene-specific counts. Subsequently, the raw counts for each gene in every sample are adjusted by dividing them by their corresponding size factors (i.e., library size adjustment), and the adjusted counts are then log-transformed. Following the normalization step, differential gene expression analysis using the negative binomial distribution by the Wald test was conducted using the DESeq function, with the wild-type group designated as the reference. The resulting list of differentially expressed genes (DEGs) was further utilized for biological pathway analysis.

2. For multiple gene comparisons in RNAseq data, p value is required to be corrected to reduce the false positive rate (FDR). However, raw P value is used in Fig 3A. Likewise, the two-sample t-test and raw P value are used in Fig 3F and 3I. A more stringent analysis would be recommended.

Reply:

We thank the reviewer for the comment. For Fig 3A, we have used Adjusted P value and have revised $-\text{Log}_{10}P$ to **$-\text{Log}_{10}P$ value (adj.)** on the Y axis. In Fig 3F and 3I, we have replaced the P value for each gene by **Adjusted P value**. Also, in the Figure 3 legends, we have replaced two-tailed unpaired student's t-test by **the Wald test**.

3. Why did authors prefer gene set score in this paper (average expression level of genes in specific gene-set)? This is not the best or most common way to evaluate pathway activity in this type of study. Over Representation Analysis (ORA), Gene Set Enrichment Analysis (GSEA) and Gene Set Variation Analysis (GSVA) may be better suited and less biased options.

Reply:

We appreciate the reviewer's thoughtful comment and agree that ORA, GSEA, and GSVA are commonly employed methods for evaluating pathway activity in gene expression studies. However, we would like to clarify that the use of gene set scores, i.e., calculating the average expression level of genes within specific gene sets, is also a widely accepted and established approach in gene expression analysis, particularly in the context of pathway enrichment analysis. Many popular packages for RNA-seq analysis, such as Seurat, provide functions to calculate gene set scores, indicating the utility and relevance of this method. The gene set scores allow us to identify pathways or gene sets that exhibit significant enrichment with differentially expressed genes, thereby providing valuable insights into the underlying biological processes driving the observed changes in gene expression. To ensure an unbiased pathway selection process, we adopted an unsupervised approach using Enrichr, a web-based tool, to identify enriched biological pathways. Subsequently, we calculated gene set scores for the pathways of interest, further highlighting their differential activity across the study groups. In the spirit of comprehensiveness, **we also supplemented our analysis with GSEA results for certain**

pathways (Figure EV3A-B), providing additional evidence and ensuring a well-rounded evaluation of pathway activity.

We are grateful for the reviewer's feedback and have considered various methods to ensure the robustness of our findings. The use of gene set scores in conjunction with unbiased pathway identification allows for a comprehensive exploration of the molecular processes underlying the observed gene expression changes. Thank you for the opportunity to address this concern, and we hope our response adequately clarifies our rationale for utilizing gene set scores in our study.

4. Three pathways in unfolded protein response are initiated simultaneously and display dissimilar dynamics under the proteotoxic stress. As XBP1s and p-EIF2a are both observed in Mitol^{-/-} HSC, why do the authors interpret that IRE1a-XBP1 pathway is the major contributor to the protective mechanism? While unfolded protein response or endoplasmic reticulum stress are clear mediators, the role of IRE1a is more tenuous. To further support this UPR/ER stress hypothesis, ATF4 protein may be a good and robust marker to measure UPR activity. If possible, Xbp1 splicing assay by qRT-PCR is recommended as an alternative method to support the IRE1a-XBP1 pathway.

Reply:

We thank the reviewer for the thoughtful and constructive comment. To address the role of IRE1a-XBP1 pathway in the protective mechanism, we examined expression of spliced Xbp1 mRNA (Xbp1s) by qRT-PCR and expression of Atf4 by flow cytometric analysis. We could confirm that **Mitol KO HSCs showed higher expression of Atf4 (Figure 5F) and Xbp1s (Figure 5G), compared to control HSCs**, which further support our hypothesis that IRE1a-XBP1 pathway contributes to UPR/ER stress in Mitol KO HSCs. These explanations were added to the result and Materials and methods section of revised text (page 10; line 248-251, page 20; line 503-507, page 21; 514-517).

5. Seahorse assay is not really "In vivo", but "ex vivo" measurement of mitochondrial respiration. It will be helpful if the authors could explain how LSK cells are collected for Seahorse assay in the Methods.

Reply:

We appreciate the reviewer's thoughtful comment. We have added the explanation regarding how LSK cells are collected for Seahorse assay in the section of Material and Methods (page 16; line 402-407). The description is shown as below,

For LSK cell sorting, firstly c-Kit⁺ cells were enriched from whole BM cells by using autoMACS Pro Separator (Miltenyl Biotec) following the treatment with magnetic beads conjugated anti-c-kit antibody (Miltenyl Biotec). After c-Kit⁺ cell-enriched fraction was then stained with corresponding antibodies for Sca1, c-Kit and Lineage markers as described above, LSK cells (Lineage⁻ Sca1⁺ c-Kit⁺) were sorted by using MoFlo XDP Cell Sorter (Beckman Coulter).

Referee #2:

Ma and colleagues provide a novel insight into the function of the outer mitochondrial membrane-localized E3 ubiquitin ligase Mitol/MArchf5 in hematopoietic stem cells (HSCs). They revealed that Mitol plays an important role in HSC function and quiescence. They nicely showed that Mitol deficient HSC have prolonged ER stress which induces IRE1a signaling. Blockage of ER stress pharmacological with KIRA6 rescues the HSC defects by reducing apoptosis.

Overall, this is a very well-written and presented manuscript describing an interesting and mechanism regulating stem cell homeostasis. The authors present their findings clearly. A real strength of this study is the identification of a novel mechanism of HSCs quiescence regulation. These findings are clearly of high relevance in the context of better understanding how to overcome stress induced apoptosis in HSC. However, there are some open questions in the study design and the current set of data as listed below.

Below are comments that aim to further increase the clarity and significance of this exciting study:

1. Can the authors please provide peripheral blood analysis from 1 week post-Poly:IC mice?

Reply:

We thank the reviewer for the constructive comment. We have added the data of peripheral blood after Poly:IC injection to **Figure EV1A**, and this explanation to the result section of revised text (page 6; line 155-156). Although the result showed the slight reduction of RBC counts in Mitol KO mice after 1-week Poly:IC injection, but WBC and platelets counts were still comparable between control and KO mice.

2. Can the authors comment whether Mitol deficient HSC in Fig.2A are able to home to the bone marrow?

Reply:

We appreciate the reviewer's comment. After control or Mitol deficient BM cells were transplanted into lethally irradiated mice, Mitol deficient BM cells showed significantly decreased chimerism in PB as well as LSK or LT-HSC fraction of BM, compared to control BM cells (Figure 2A-C). Although similar tendency was observed when Mitol deletion was induced after generated BM-chimeric mice (Figure 2D-G), this suppressive effect seems to be smaller compared to that observed after the transplantation. These data support that Mitol deficiency negatively affects HSC maintenance, but simultaneously suggest the possibility that homing capacity is also affected by Mitol deletion.

3. In Fig.4C the authors measured MitoSox levels in Mitol deficient mice, however the histograms show a bimodal distribution of the dye. The MitoSox high peak is even smaller in the knockout mice. Can the authors comment on this finding?

Reply:

We thank the reviewer for the comment. MitoSox was used to measure mitochondrial superoxide production in living cells. We expect that the bimodal distribution in the histograms may be attributed to the heterogeneity of LT- and ST-HSC fraction. Indeed, several group including us showed the functional, epigenetic or metabolic heterogeneity within HSC fraction (Giladi A et al., *Nat Cell Biol*, 2018, 20: 836-846; Wilson NK et al., 2015, 16: 712-24; Mansell E et al., *Cell Stem Cell*, 2021, 28(2):241-256.e6; Umemoto et al., *EMBO J*, 2022, 41: e109463). Due to such heterogeneities, each cell within these cell fractions including LT-HSCs may exhibit variable metabolic status, resulting in a bimodal distribution of the dye.

4. The authors should provide more insights in difference in glucose uptake by using 2-NBDG in wildtype vs. Mitol-deficient mice.

Reply:

We thank reviewer for the comment. To address this, we examined the potential for glucose uptake, and confirmed **little difference between control and Mitol KO HSCs as shown in Figure 4E**. These results suggested that Mitol^{ΔΔ} HSPCs showed increased glycolysis without relying on significantly enhanced glucose uptake. This result is consistent with gene expression pattern related to glycolysis (Fig EV3A). However, Mitol deletion enhanced ECAR in HSPCs (Fig 4G). These findings may suggest that Mitol deletion enhances glycolysis without enhanced glucose uptake, but the mechanism is still unknown mechanism. We added these explanations and accompanied correction into the revised text (page 9; line 224-225 and 231-232, page 11-12; line 284-295, page 20; line 495-501).

5. The authors should also provide Seahorse data from 4 week post-Poly:IC mice to better understand the metabolic adaptations of Mitol deficient LSK cells.

Reply:

We appreciate the reviewer's thoughtful comment. However, we found that after 4 weeks post Poly:IC injection, BM cellularity was greatly decreased, which could affect the metabolic status of HSCs in Mitol^{ΔΔ} mice through a change in the BM environment. Thus, it is too challenging to clearly examine the effect of Mitol deletion on mitochondrial metabolism after 4 week from Poly:IC injection due to the large difference of BM environment between control and Mitol^{ΔΔ} mice. Instead, little difference in nucleated cell number within the BM is observed between control and KO mice, at least in 1 week post Poly:IC injection samples, suggesting that Mitol deficiency does not affect BM environment yet.

Therefore, we focused on 1 week rather than 4 weeks as the appropriate timing to more clearly examine the effect of Mitol deletion on metabolic status in HSCs.

6. The authors nicely show that IRE1a inhibition with KIRA6 rescues the phenotype in vitro. The authors should show the same effect in vivo treating mice with KIRA6 (PMID: 25018104).

Reply:

We thank the reviewer for the thoughtful and constructive comment. To address this, we serially administered KIRA6 into control or Mitol KO mice following Poly:IC injection (Figure A for reviewers). The administration of KIRA6 recovered HSC frequency within BM of KO mice close to that in control mice (Figure B for reviewers). However, the similar positive effect of KIRA6 were observed in the frequency of not only HSC but also LEK fraction within BM of control mice. Due to such an effect of KIRA6, it remains unclear whether KIRA6 rescue from decreased HSC frequency in Mitol KO mice through suppressing ER stress. Thus, it is difficult to clearly examine the relationship between the negative effect of Mitol deficiency on HSCs and ER stress *in vivo* by using KIRA6.

On the other hand, we observed that KIRA6 could not fully rescue from Mitol KO-mediated increased frequency of ST-HSCs or MMPs (Figure B for reviewers). Based on our RNA-seq data and pathway analysis, as the physiological conditions caused by Mitol deficiency are extremely complicated (Figure 3B-C), we suggest that Mitol deficiency exerts multi-factor influences in addition to ER stress *in vivo*.

Figure for reviewers: KIRA6 administration *in vivo*.

(A) Schematic figure of KIRA6 administration *in vivo*.

(B) HSPC population distribution after 1 week of KIRA6 administration (10 mg/kg/day) following 1 week of Poly:IC injection (n= 3-4). Data represent mean \pm SD with two-tailed unpaired student's *t*-test. ns, $P > 0.05$; *, $P < 0.05$; **, $P < 0.01$; ***, $P < 0.01$.

Dear Toshio, dear Terumasa,

Thank you for submitting your revised manuscript (EMBOJ-2023-114405R) to The EMBO Journal. As mentioned, your amended study was sent back to the referees for their re-evaluation, and we have received comments from both of them, which I enclose below. As you will see, the experts state that the work has been substantially improved by the revisions and they are now broadly in favour of publication.

Thus, we are pleased to inform you that your manuscript has been accepted in principle for publication in The EMBO Journal.

Also, we now need you to take care of a number of minor issues related to formatting and data presentation as detailed below, which should be addressed at re-submission.

Please contact me at any time if you have additional questions related to below points.

As you might remember from previous experience, every paper at the EMBO Journal now includes a 'Synopsis', displayed on the html and freely accessible to all readers. The synopsis includes a 'model' figure as well as 2-5 one-short-sentence bullet points that summarize the article. I would appreciate if you could provide this figure and the bullet points.

Thank you for giving us the chance to consider your manuscript for The EMBO Journal. I look forward to your final revision.

Again, please contact me at any time if you need any help or have further questions.

Best regards,

Daniel

>> Adjust the title of the 'Conflict of Interests' section to 'Disclosure and Competing Interests Statement'. Please add a sentence referring to your advisory board membership (T.S.), 'T.S. is member of the EMBO Journal editorial advisory board.'

>> Author Contributions: Please remove the author contributions information from the manuscript text. Note that CRediT has replaced the traditional author contributions section as of now because it offers a systematic machine-readable author contributions format that allows for more effective research assessment. and use the free text boxes beneath each contributing author's name to add specific details on the author's contribution.

More information is available in our guide to authors.

>> Please adjust the reference format to EMBO Journal style, limiting to 10 authors et al. . DOIs for published journal articles should be removed.

>> Dataset EV legends: Please rename "Table EV1" and "Table EV2", also in callouts; upload as two separate files.

>> Data availability section: remove the referee token and make sure privacy is released for the GEO dataset.

>> Consider additional changes and comments from our production team as indicated below:

-Figure legends:

1. Please note that a separate 'Data Information' section is required in the legends of all the figures.

2. Please indicate the statistical test used for data analysis in the legends of figures EV2a-d; EV3a-b; EV4a

3. Please note that the box plots need to be defined in terms of minima, maxima, centre, bounds of box and whiskers, and percentile in the legends of figures 3f, i; EV3c-e; EV5d

4. Please note that the error bars are not defined in the legends of figures 3d, g; EV4a; EV5a, c

5. Please note that information related to n is missing in the legends of figures 4b, d, f, h; 5a, c; EV1e; EV3c-f; EV4a; EV5a, c, d
6. Please note that the scale bar needs to be defined for figure 5a

We realize that it is difficult to revise to a specific deadline. In the interest of protecting the conceptual advance provided by the work, we recommend a revision within 3 months (11th Mar 2024). Please discuss the revision progress ahead of this time with the editor if you require more time to complete the revisions.

Referee #1:

The authors have done a great job revising the manuscript and I have no further comments.

Referee #2:

This is a revised version of the previously submitted manuscript from Ma and colleagues. The authors have performed a thorough revision of the manuscript and included extended series of novel interesting observations and data.

The authors have fully addressed all of my comments, and therefore I recommend this exciting highly clinically relevant work to be accepted for publication.

The authors addressed the minor editorial issues.

Dear Toshio, dear Terumasa,

Thank you for submitting the revised version of your manuscript. I have now evaluated your amended manuscript and concluded that the remaining minor concerns have been sufficiently addressed.

Thus, I am pleased to inform you that your manuscript has been accepted for publication in the EMBO Journal.

Your manuscript will now be processed for publication by EMBO Press. It will be copy edited and you will receive page proofs prior to publication. Please note that you will be contacted by Springer Nature Author Services to complete licensing and payment information.

Please note that it is The EMBO Journal policy for the transcript of the editorial process (containing referee reports and your response letter) to be published as an online supplement to each paper. I would accordingly like to ask your consent on keeping the referee figure included in this file.

If you do NOT want the transparent process file published, you will need to inform the Editorial Office via email immediately. More information is available here: https://www.embopress.org/transparent-process#Review_Process

On a different note, I would like to alert you that EMBO Press offers a format for a video-synopsis of work published with us, which essentially is a short, author-generated film explaining the core findings in hand drawings, and, as we believe, can be very useful to increase visibility of the work. This has proven to offer a nice opportunity for exposure i.p. for the first author(s) of the study. Please see the following link for representative examples and their integration into the article web page:

<https://www.embopress.org/doi/full/10.15252/embo.2019103932>

If you have any questions, please do not hesitate to call or email the Editorial Office.

with
Best regards,

Daniel

Daniel Klimmeck, PhD
Senior Editor
The EMBO Journal
EMBO
Postfach 1022-40
Meyerohofstrasse 1
D-69117 Heidelberg
contact@embojournal.org
Submit at: <http://emboj.msubmit.net>